# Poly (N-Vinylcaprolactam-Grafted-Sodium Alginate) Based Injectable pH/Thermo Responsive In Situ Forming Depot Hydrogels for Prolonged Controlled Anticancer Drug Delivery; In Vitro, In Vivo Characterization and Toxicity Evaluation

**DOI:** 10.3390/pharmaceutics14051050

**Published:** 2022-05-13

**Authors:** Samiullah Khan, Muhammad Usman Minhas, Muhammad Tahir Aqeel, Ihsan Shah, Shahzeb Khan, Mohsin Kazi, Zachary N. Warnken

**Affiliations:** 1Margalla College of Pharmacy, Margalla Institute of Health Sciences, Rawalpindi 46000, Punjab, Pakistan; tahiraqeelmalik@gmail.com (M.T.A.); ihsanshah8725@gmail.com (I.S.); 2College of Pharmacy, University of Sargodha, Sargodha 40100, Punjab, Pakistan; us.minhas@hotmail.com; 3Department of Pharmacy, University of Malakand, Chakdara 18800, Khyber Pakhtunkhwa, Pakistan; 4Discipline of Pharmaceutical Sciences, School of Health Sciences, UKZN, Durban 4041, South Africa; 5Department of Pharmaceutics, College of Pharmacy, King Saud University, P.O. Box 2457, Riyadh 11451, Saudi Arabia; mkazi@ksu.edu.sa; 6Division of Molecular Pharmaceutics and Drug Delivery, Department of Molecular Pharmaceutics and Drug Delivery, College of Pharmacy, The University of Texas at Austin, Austin, TX 78712, USA; zwarnken@utexas.edu

**Keywords:** chemical grafting, N-(vinylcaprolactam), hydrogels, sodium alginate, 5-FU in situ depot, rheology, MTT assay, pharmacokinetics, anticancer drugs

## Abstract

This study was aimed to develop novel in situ forming gels based on N-vinylcaprolactam, sodium alginate, and N,N-methylenebisacrylamide. The in situ Poly (NVRCL-g-NaAlg) gels were developed using the cold and free radical polymerization method. The structure formation, thermal stability, and porous nature of gels was confirmed by FTIR, NMR, DSC, TGA, and SEM. The tunable gelation temperature was evaluated by tube titling and rheological analysis. Optical transmittance showed that all formulations demonstrated phase transition around 33 °C. The swelling and release profile showed that gels offered maximum swelling and controlled 5-FU release at 25 °C and pH (7.4), owing to a relaxed state. Porosity and mesh size showed an effect on swelling and drug release. The in vitro degradation profile demonstrated a controlled degradation rate. An MTT assay confirmed that formulations are safe tested against Vero cells. In vitro cytotoxicity showed that 5-FU loaded gels have controlled cytotoxic potential against HeLa and MCF-7 cells (IC_50_ = 39.91 µg/mL and 46.82 µg/mL) compared to free 5-FU (IC_50_ = 50.52 µg/mL and 53.58 µg/mL). Histopathological study demonstrated no harmful effects of gels on major organs. The in vivo bioavailability in rabbits showed a controlled release in gel form (C_max_, 1433.59 ± 45.09 ng/mL) compared to a free drug (C_max_, 2263.31 ± 13.36 ng/mL) after the subcutaneous injection.

## 1. Introduction

Improving the quality of health care and treatment of a disease require the development and applications of novel biomaterials. Current research in the field of biomaterials focuses on the synthesis of new materials with high biocompatibility, responsiveness to internal and external environmental stimuli, and enhanced mechanical properties. Polymeric biomaterials have wide applications, specifically in modified drug delivery systems, artificial organs, dental implants, tissue engineering, stents coated with polymer, and coated tablets [1,2].

The realization of therapeutic efficacy of a therapy depends upon the release of the drug from a delivery system that plays an important role in the development of time-controlled devices. So, the optimization of the pharmacotherapy demands that the release of the drug moiety from a delivery system should be in accordance with the therapeutic goals and the pharmacological possessions of the drug. Moreover, the delivery system needs to be effectively modified therapeutically for the elimination of potential risks that may arise due to under dose or over dose [3].

Hydrogels, a class of biomaterials with 3D structure, have the ability to swell and shrink as a result of diffusion of aqueous or biologic media in and out of the network, respectively, which is the most important characteristic of hydrogels [4].

In the field of timed or controlled release devices, environmental responsive hydrogels have gained considerable interest, which have the ability to react to a minute change in the external or internal stimuli [5]. The stimuli to which these smart monomers or polymers show response mainly include pH, temperature, light, ionic strength, electric field, magnetic field, and type of salt. These responsive matrices are also called “materials with brain” due to their intelligent characteristics. These responsive matrices have close resemblance with the natural physiological processes, in which the release of encapsulated drug can be modulated by the change in physiological condition [6].

In general, most of the polymers show solubility in aqueous media on heating, but some water-soluble polymers make precipitates from the solution on heating. This unique characteristic is possessed by the group of polymers that become soluble on cooling and phase separate on heating at a certain temperature point, which is known as lower critical solution temperature (LCST) [7]. At this temperature, the thermoresponsive monomer or polymer undergoes a sharp coil-to-globule transition. This phase transition or reversible conformational changes for thermoresponsive polymers occur in response to a small variation in temperature [8]. Below LCST, the hydrophilic state remains in dominant form in which the water molecules remain attached with the network chain; while above LCST, the hydrophobic interactions become dominant with the release of water molecules from the network chain.

Scientists in the field of biomaterials are now twisting their focus towards the biodegradable injectable hydrogels that serves as potential materials for delivery of bioactive agents and proteins as well as used in the field of tissue engineering. These injectable sol–gel transition hydrogels that are based on thermoresponsive polymers act as an in situ depot system at physiological temperature for the controlled delivery of encapsulated therapeutic agents upon injection into the body [9,10,11].

Generally, for the synthesis of in situ injectable depot hydrogels, physical or self-assembled and chemical crosslinking approach are used. Physically cross-linked hydrogels, which are formulated from the self-assembly of a wide variety of molecules, utilize weak interaction forces such as hydrophobic interactions, hydrogen bonding, electrostatic interactions, host/guest inclusion complex, stereocomplex formation, and π−π staking. These physical or reversible crosslinks permit the design of “smart” materials that have the ability to self-adapt specifically in response to a change in environment (in terms of temperature, pH, ionic strength, electric field, etc.) [12]. Chemically cross-linked gels, which are referred to as permanent gels, are formed by the covalent crosslinking between reacting moieties. This covalent crosslinking is formed with the use of crosslinking agents such as glutaraldehyde, dialdehyde, epoxy compounds, and formaldehyde [13].

In the design of controlled drug delivery devices, carbohydrate polymers have widely been investigated in pharmaceutical and biomedical fields. The polysaccharides, which are less expensive, biodegradable, biocompatible, nontoxic, freely available, and renewable, do hold great advantages over their synthetic counterparts [14]. Polysaccharides, such as chitosan, xanthan gum, gellan gum, pectin, guar gum, and sodium alginate, have been used either alone or in combine modified form with their native moieties to control the release of bioactive agents from different types of delivery devices [15,16].

Alginate composed of two monomeric units, β-D-mannuronic acid (M) and α-L- guluronic acid (G), is linear anionic polysaccharide obtained from natural source (brown algae) belonging to the phaeophyceae. Sodium alginate (NaAlg) is a biodegradable, renewable natural polymer that not only has applications in the design of controlled release devices but also is used as a colloidal stabilizer and thickening agent, as well asin gelatinization [17]. The sodium alginate has, however, some inherent disadvantages of poor mechanical strength. In order to overcome this problem, chemical modifications via graft copolymerization have been previously reported by some researchers with some synthetic monomers. Grafting of NaAlg by vinyl monomers is considered to be one of the most successful approaches that provide the additional properties maintaining the biodegradable nature of the backbone. Previously pH/thermo dual responsive graft copolymer based on NaAlg has been reported to make them attractive biomaterials in controlled release applications [18,19].

N-(vinylcaprolactam) (NVCL) is a biocompatible, nonionic, thermal sensitive, nontoxic, hydrophilic, and water-soluble monomer at room temperature [20]. Moreover, NVCL possesses lower critical solution temperature (LCST) in the physiological range (32–38 °C) that has widened its applications in pharmaceutical industry [21]. This monomer contains carboxylic and amide groups in its structure, where the amide group is directly connected to the hydrophobic carbon-carbon backbone chain [22]. The LCST of the NVCL has a direct connection with its molecular weight, and the concentration of the monomer [23] NVCL solutions and gels has considerable interest of scientists due to its water-solubility, biocompatible nature, non-adhesiveness, and, reversely, thermoresponsive behavior near the body temperature [24]. Presently, there are no studies reporting the in situ gelling depot formulation for drug delivery applications of thermoresponsive block copolymer based on NVCL grafted with NaAlg.

Cancer is a major cause of human deaths nowadays, and chemotherapy has remained a major approach of treatment for malignant tumors [25]. The success of chemotherapy is limited by undesirable severe toxic effects, chemoresistance, low tumor uptake, and enhanced diffusion to normal healthy cells [26]. The delivery of the chemotherapeutic drugs to the tumor tissues with minimally diffusion to other organs or tissues is the compelling alternative to overcome the limitations associated with the chemotherapy. 5-Flourouracil (5-FU) is one of the best and oldest chemo drugs that is commonly used against many types of tumors in clinical practice, i.e., rectal, breast, ovarian, colon, liver, gastrointestinal, head, and neck cancers. Current treatment modalities with this drug trigger some limitations such as short biological half-life due to rapid metabolism by dihydropyramidine dehydrogenase, non-uniform, and incomplete oral absorption and toxicity against healthy cells due to the non-selective action [27]. In order to reduce the toxicity and increase the therapeutic efficacy, some research groups have developed and report various drug delivery devices including coated micro and nanoparticles, liposomes, polymeric microspheres, emulsions, polymeric micelles, and various other prodrug systems including polymer-drug conjugates [28,29,30].

An injectable in situ gelling depot delivery system is a new approach that can effectively sustain the drug release, which may lead to improved patient compliance and efficiency with the subsequent less toxic effects on other tissues and organs.

Our group is working on the design of stimuli responsive injectable depot hydrogels for the delivery of different pharmaceutical agents. In this study, series of chemically grafted thermoresponsive injectable in situ depot hydrogels sodium alginate were prepared via cold and free radical polymerization techniques. To the best of our knowledge and in the literature, no such detailed study has been conducted to date. Physical characteristics including clarity of the gels, rheological measurement, sol-gel phase transition, optical transmittance, gelation temperature, and gelation time were determined for the developed chemically grafted depot gels. The developed depot gels were subjected to swelling tests, on–off switching behavior, drug content determination, and in vitro drug release experiments in various swelling medias and at different temperatures. The developed formulations were subjected to degradation to investigate the in vitro degradation profile. For chemically grafted depot gels, different networking parameters, i.e., diffusion coefficient (D), volume fraction of polymer (V_2s_), solvent interaction parameters (χ), and average molecular weight (Mc) were calculated using the Flory–Huggins theory. A methyl thiazolyl tetrazolium (MTT) assay was used to study the safety and toxic potential of blank and drug loaded gels against Vero cell lines as well as Hela and Breast cancer cell lines (MCF-7 cells). The copolymer structure formation and thermal properties were assessed via nuclear magnetic resonance (NMR), fourier transform infrared (FTIR), differential scanning calorimetry (DSC), and thermogravimetric analysis (TGA), while the porous gels structure was assessed via scanning electron microscopy (SEM).

## 2. Materials and Methods

### 2.1. Materials

All materials were reagent grade and obtained from Aldrich unless otherwise noted. N-(Vinylcaprolactam, MW = 139.19 g/mol) (98%, Sigma Aldrich, Steinheim am Albuch Germany) was used without further treatment and stored at 4°C. Sodium Alginate (NaAlg) (MW = 10,000–600,000 Da) was purchased from Daejung Chemical Company Seoul, Korea. N,N-Methylene-bis-acrylamide (MBA) was purchased from Fluka, Switzerland. 5-Flourouracil (Purity 99%) was purchased from Sigma Aldrich, Steinheim am Albuch Germany. Ammonium persulphate (NH_4_)_2_SO_4_, APS, and Acetic acid were purchased from Sigma Aldrich, Steinheim am Albuch Germany. Sodium dihydrogen phosphates, sodium hydroxide, and NaCl were purchased from from Daejung Chemical Company Seoul, Korea. Deionized water was used for conducting experiments. Minimum Essential Medium added with penicillin, l-glutamine, and streptomycin containing fetal bovine serum (FBS) were obtained from San francisco, USA, Invitrogen.

### 2.2. Preparation of Chemically Grafted Thermoresponsive Poly (N-Vinylcaprolactam-Graft-Sodium Alginate) In Situ Depot Gels

Thermoresponsive chemically grafted Poly (N-Vinylcaprolactam-graft-Sodium Alginate) i.e., Poly (NVCL-g-NaAlg) in situ gels with various ratios of polymer, monomer, drug, and cross linker were prepared via a combination of cold and free radical polymerization method as shown in Table 1. A briefly weighed amount of N-(Vinylcaprolactam) was dissolved at 300 RPM in cold distilled water for 1 h and stored at 4 °C for 24 h until a transparent solution obtained. Then, NaAlg in a weighed amount was dissolved in distilled water at room temperature and 300 RPM for 30 min. The previously prepared N-(Vinylcaprolactam) solution was then added drop wise to an NaAlg solution with continuous stirring. A weighed amount of ammonium persulphate (1.5% *W*/*W*) was added to the copolymer solution to initiate the solution polymerization, and the dispersion was kept at stirring till complete mixing. N,N-methylenebisacrylamide (MBA) was dissolved in distilled water and added to a previously prepared copolymer mixture drop wise. The final co-mixture was filtered (0.45 micron), incorporated 5-FU (10 mg/mL), and kept on stirring for 1 h under nitrogen bubbling. The final prepared mixture was stored at 4 °C and assessed for clarity, phase transition, rheology, and other characterization [17]. Appendix A in the Appendix A refers to the proposed Poly (NVCL-g-NaAlg) gels structure.

### 2.3. Clarity of the In Situ Gel Formulations

The clarity of in situ thermosensitive polymeric solutions and chemically grafted formed gel was observed visually at various temperature values, i.e., 4 °C, 25 °C, and 37 °C [31].

### 2.4. Solid State Characterization of Gels

#### 2.4.1. ^1^H Nuclear Magnetic Resonance (NMR) Spectroscopic Analysis

To confirm the monomer, polymer, and grafted copolymer structure, samples were subjected to ^1^H-NMR (400 plus spectrometer, Bruker Ltd., Coventry, UK) analysis using deuterium oxide (D_2_O) (as a solvent) and tetramethylsilane (TMS) as internal standard at a frequency of 400 MHz. The chemical shifts were represented in parts per million downfield (ppm) [32].

#### 2.4.2. FTIR Analysis

The FT-IR spectra of pure NaAlg, 5-FU, NVCL, and Poly (NVCL-g-NaAlg) gel sample were recorded for possible interactions using FT/IR-4100 Series (Jasco, UK) from 4000 to 600 cm^−^^1^, with a resolution of 8.0 cm^−^^1^ at room temperature equipped with MIRacle^TM^ software (9.0 version) [33].

#### 2.4.3. Thermogravimetric Analysis (TGA)

Thermogravimetric analyzer (Model: DTG 500, Shimazdu, Japan) was used to record the decomposition thermograms of selected samples from 25 °C to 300 °C at 10 °C/min heating rate and 30 mL min^−^^1^ nitrogen gas flow rate [34].

#### 2.4.4. Differential Scanning Calorimetry (DSC)

TA instruments Q100 DSC thermal apparatus was used to analyze the thermal stability and behavior of pure materials and chemically grafted gel samples. Selected samples (5 mg) were sealed in an aluminum pan and heated between 20–300 °C under N_2_ purging at 20 °C/min [17].

#### 2.4.5. Scanning Electron Microscopy (SEM)

The surface and internal morphologies of Poly (NVCL-g-NaAlg) gels were assessed using JEOL SEM (JSM-6490A, Tokyo, Japan). Dried powdered samples were placed on an aluminum stub using adhesive tape. Then, gold was coated on the stubs using gold sputter under argon atmosphere. The Surface and internal structures of samples were analyzed by taking photomicrographs at 20 KV voltage and chamber pressure of 0.6 mm Hg [35,36].

### 2.5. Cloud Point Determination by Titling Method (Tsol-Gel)

The phase change temperature (Tsol-gel) of optimized samples reported in Table 1 was determined by tube titling method [31]. Each time, 5 mL sample of final co-polymer solution was added to glass tube and wrapped with parafilm. The glass tube was retained initially at 20 °C in digital water bath and heated slowly at 1 °C/min, to the temperature at which the mixture did not move on inverting at 90 degrees. For gelation time measurement, briefly, the copolymer sample in glass tube was placed in a water bath at 32 °C, and sample flow was noted every 10 s by inverting the tubes. The time point at which samples stopped flowing was recorded as the gelation time.

### 2.6. Rheological Determination

The thixotropic nature of Poly (NVCL-g-NaAlg) depot gels was assessed by monitoring viscosity, storage modulus (G′), and loss modulus (G″) using AR-2000 rheometer fitted with 40 mm parallel plate and circulating bath. Time sweep tests were conducted at 30 °C and an oscillatory frequency of 1 rad/s to investigate the gelation kinetics with time. Temperature sweep tests were performed to monitor temperature dependent viscoelastic properties by changing the bath temperature over 25–40 °C at 2 °C per minute. The effect of temperature on viscosity, G′, and G″ in oscillation mode was evaluated at 1 rad/s frequency and 0.1 Pa shear rate. The change in viscosity along increasing shearing rate (0.1–10 s^−^^1^) was examined in flow mode.

For stability determination of the gels at LCST (33 °C), samples were scrutinized by frequency sweep test. The stability of samples was monitored by noting change in G′ and G″ at 0.01–50 Hz frequency range at 1% strain [37].

### 2.7. Optical Transmittance Measurement

The optical absorbance’s of Poly (NVCL-g-NaAlg) gels were measured at 450 nm and varied temperatures using UV-Visible spectrophotometer. The transmittances were measured at digital water bath under 25–45 °C range using 10 mm × 10 mm × 40 mm disposable cuvettes. Prior to transmittance analysis, every test sample was kept at that temperature for 5 min [37].

### 2.8. Swelling Experiments

The swelling profile of blank chemically grafted Poly (NVCL-g-NaAlg) depot gels was analyzed in distilled water (DW), pH 7.4 (PBS, 5 mM) and pH 1.2 (0.1 M HCl) at different temperatures in closed containers. For Chemically grafted Poly (NVCL-g-NaAlg) in situ depot gels, the swelling experiments were conducted at 25°C and 37 °C. At definite time interims, the swollen samples were taken out, blotted surface with soft paper, weighed, and placed back in solution until equilibrium weight was achieved. The swelling ratio was calculated using following formula [38];
(1)Swelling Ratio (SR)=Ws−WdWd
where *W_s_* and *W_d_* indicates the weight of hydrogel samples in swollen and dry states, respectively.

#### 2.8.1. On–Off Switching Behavior

The on–off switching behavior (oscillatory swelling-deswelling-reswelling cycles) of blank chemically grafted Poly (NVCL-g-NaAlg) hydrogels was studied in PBS (5 mM) of pH 7.4 and temperature near around the phase transition temperature. Briefly, the gel sample was placed in PBS for 2 h at 25 °C and then transferred to a vessel containing PBS at 37 °C for 2 h, and this cycle was continuously run. Each time, swollen disc weight was recorded prior to alternate immersion. The SR was measured by using Equation (1) [39,40].

#### 2.8.2. Solvent Diffusion Coefficient

Release of the drug from chemically grafted hydrogels generally follows a diffusion mechanism. Diffusion coefficient (*D*) refers to substance diffusion with concentration gradient across unit area in unit time. For “*D*” values determination, dried samples were placed in PBS (pH = 7.4) owing to their greater swelling. “*D*” values of gels were determined by the following formula [41].
(2)D=πh.θ4.Qeq2
where *D* denotes the diffusion coefficient, *Q**_eq_* mentions the equilibrium swelling degree of gels, 𝜃 is the slope, and refers to sample disc thickness in dry state.

#### 2.8.3. Molecular Weight between Cross-Links (Mc)

*Mc* values of Poly (NVCL-g-NaAlg) gels were determined using the Flory–Rehner theory. For *Mc* values calculation, the following equation was used [41].
(3)Mc=−dpvsv2,s1/3−v2,s2ln1−v2,s+v2,s+xv2,s2

#### 2.8.4. Volume Fraction of Polymers

Polymers volume fraction (*V*_2,*s*_) of hydrogels indicates the absorbing ability of gels in its porous structure. For calculating (*V*_2,*s*_)of samples, the following formula was used;
(4)V2,s=1+dhdsMaMb−1−1
where *d_h_* and ds designate the densities (g/mL) of gel sample and solvent, respectively. *M_a_* and *M_b_* denote masses (g) of the samples in swollen and dry states [41]. 

#### 2.8.5. Solvent Interaction Parameters (*χ*)

Solvent interaction parameter was used to evaluate the gels components compatibility with surrounding fluids. For (*χ*) values determination the Flory–Huggins theory was used by applying following equation [41].
(5)χ=ln(1−ν2,s)+ν2,sν2,s2
where *V*_2,*s*_ (mL/mol) represents volume fraction of gels in swollen state.

### 2.9. Percent Crosslinking Measurements

For percent crosslinking, dried gel samples were preserved in distilled water (50 mL) under stirring for 24 h to remove the uncrosslinked portions. After 24 h, samples were taken out and, then, dried at 37 °C till uniform weight was achieved. Percent crosslinking was measured by the following formula [42];
(6)Percent Crosslinking=W2W1×100
where *W*_1_ and *W*_2_ designate the weights of dried discs before and after experiment.

### 2.10. Drug Contents Determination

Extraction method was used for the measurement of loaded drug from Poly (NVCL-g-NaAlg) in situ gels. A total of 10 g of the 5-FU loaded sample (100 mg/g) was cut into samples of 7 mm. Samples were dipped in 25 mL of pH 7.4 (PBS, 5 mM) for 24 h used as extracted solvent. Drug contents extracted were determined at 265 nm using UV-Visible spectrophotometer [43,44]. The % drug loading was determined using the following formula;
(7)Drug loading %=WT−WSWT×100. 
where *W_T_* is the total drug content in the gels and *W_S_* is the drug contents in the supernatant.

### 2.11. Grafting Efficiency Measurement

The grafting of NaAlg to NVCL was assessed by the procedure reported by Zhang L et al. [44]. The efficiency and grafting percentage were calculated by noting the weight difference before and after reactions using the formula;
(8)Grafting Efficiency % (GE)=Wg−WNaAlgWNVCL×100
where *W_g_*, *W_NaAlg_*, *W_NVCL_* are the weights of purified grafted copolymer, sodium alginate (*NaAlg*), and *NVCL*, respectively.

### 2.12. In Vitro Drug Release

The in vitro drug release profile of Poly (NVCL-g-NaAlg) gels was observed at 25 °C and 37 °C using USP dissolution apparatus-II in distilled water, pH 7.4 (PBS, 5 mM), and pH 1.2 (0.1 M HCl), respectively. The drug-encapsulated samples were placed at desired pH and selected temperature. At regular time interim, 2 mL aliquot was withdrawn and analyzed for release using UV-Visible spectrophotometer at 265 nm. The sink conditions were maintained by adding fresh media [37,40]. The cumulative % release was calculated by using following formula;
(9)% Drug Release=50Cn+2Σ Cn−1m0×100. 
where: 50 designates volume of medium in ml; Cn and Cn−1 refers to the concentration after n and n − 1 times, respectively; 2 refers to the withdrawn volume for analysis in ml; and m_0_ designates the drug amount (mg) loaded in gels.

### 2.13. Drug Release Kinetics

Drug release kinetics were assessed by fitting the in vitro experimental data obtained to various mathematical models [31,40].

#### 2.13.1. Zero Order Kinetic Model

Qt shows the amount of drug dissolved in time t, Q_0_ indicates the initial drug amount in the solution, and K_0_ is zero order constant.
Q_t_ = Q_0_ + K_0_ t(10)

#### 2.13.2. First Order Kinetic Model

Mo indicates the initial amount of drug, Mt indicates the remaining drug amount at time t, and k_1_ is the first order constant.
ln Mt = −k_1_t + lnMo (11)

#### 2.13.3. Higuchi Model

The model relates cumulative drug release versus square root of time as shown in Equation (12)
M = kH t_1/2_(12)
where M indicates the amount of drug released at time t, and kH is the Higuchi rate constant.

#### 2.13.4. Korsmeyer–Peppas Model

This model exponentially relates the drug release to the elapsed time. The equation is given as
Ln (Mt/M∞) = ln kp + n ln t(13)

(Mt/M∞) is the fraction of drug released at time t and n is the slope, which determines the type of diffusion from the polymer matrix.

### 2.14. In Vitro Degradation

A weight loss method was adopted for monitoring in vitro degradation profile of Poly (NVCL-g-NaAlg) hydrogels. Briefly, each sample was initially weighed and incubated in container containing PBS (pH = 7.4, 10 mL) at 40 rpm using shaking incubator at 37 °C with continues PBS replenishment. At definite time interims, samples were rapidly lyophilized at −80°C and weighed again. The % weight loss was calculated using the following formula [32];
(14)Weight Loss (%)=W0−WtW0×100
where *W*_0_ and *W_t_* refers to sample weights before and after degradation, respectively.

### 2.15. Cell Lines and Culture Conditions

Human cervical (Hela cells), breast (MCF-7) cancer cell lines, and Vero cell lines obtained from Cells culture collection laboratory, Noori Cancer Hospital, Islamabad, Pakistan, were cultured in Minimum Essential Medium (MEM) and stored in an incubator at 37 °C provided with 5% CO_2_. At 90% confluency, the cells were separated with Trypsin-EDTA, centrifuged for 3 min at 3000 rpm, and, then, resuspended in medium for further use.

Methyl thiazolyl tetrazolium (MTT) assay. The cells’ compatibility and toxicity with blank and drug loaded gels were evaluated via MTT assay. Vero cells were used to investigate the compatibility of blank gels, while Human cervical (HeLa) and breast adenocarcinoma cancer cell line (MCF-7) were used to evaluate the anticancer potential of 5-FU solution and 5-FU loaded chemically grafted Poly (NVCL-g-NaAlg) hydrogels by standard MTT assay. Cells were incubated in 96 well plate and, then, free 5-FU solution (100 µL) and 5-FU containing hydrogel solution (100 µL) were injected into the wells as per concentration defined (1, 5, 10, 20, 40, 60, 80, 100 μg/mL). The plates were then incubated for 24 h at 37 ± 1 °C. Untreated wells alone were considered as negative control, and wells applied with Triton X100 and free 5-FU were used as positive control. The cells’ viability was investigated by colorimetric determination using ELISA reader and the assay was terminated after 24 h incubation. The cells’ viability was measured at 570 nm on Bio-Rad 680 microplate reader. Cells’ viability (%) was measured by using following formula [34,37];
(15)Cells Viability %=AsampleAcontrol×100
where *A_sample_* and *A_control_* are the absorbances of the sample and control wells, respectively.

### 2.16. In Vivo Analysis in Rabbits

#### 2.16.1. High Performance Liquid Chromatography Analysis

A simple, accurate, sensitive, and reproducible HPLC method previously developed was validated and used for 5-FU analysis in rabbit’s plasma. The absorbed amount of 5-FU in plasma samples was quantified using an HPLC system coupled with BDS hypersil C18 stainless steel column (5 µm, 4.6 mm × 250 mm) at detection wavelength of 265 nm. The mobile phase was consisted of filtered double distilled water adjusted to pH 3.2, with orthophosphoric acid and acetonitrile (70:30) injected and eluted at flow rate of 1 mL/minute.

#### 2.16.2. Animal Handling

Animal models consisted of healthy albino rabbits (2.0 to 2.5 kg) were selected and obtained from the animal section of Pharmacology Research Laboratory, Faculty of Pharmacy, the Islamia University of Bahawalpur, Pakistan. The Pharmacy Research Ethics Committee (PREC) has reviewed and approved the study protocols. Rabbits were kept under well-controlled room conditions (25 ± 1 °C) individually in wooden boxes, fed with commercial laboratory rabbit diet, and freely allowed to water before the initiation of study.

#### 2.16.3. Drug Administration and Sampling

In order to administer drug, a total of 12 rabbits were randomly divided into two groups (group A and group B), having 6 rabbits in each group, and were placed in wooden boxes. For identification purpose during handling of drug administration and sampling process, all rabbits were tagged properly. Before dosing, all the rabbits were restricted from food and were kept fasted for 12 h, but given free access to water. The study was carried out on the basis of parallel study design. In the first phase, free drug solution (5-FU, 20 mg/2 mL) was injected subcutaneously to group A (control group). In the second phase, group B was injected with 5-FU loaded selected thermoresponsive copolymeric solution (2 mL containing 20 mg of 5-FU) through the subcutaneous route. Before 5 min of injecting the formulation and at specific time intervals, blood samples (500 µL) were collected from marginal ear vein of each rabbit in heparinized tubes. The plasma was separated by centrifugation at 1600× *g* for 10 min and, then, stored at −80 °C till analysis in an ultra-low freezer (Sanyo, Japan).

#### 2.16.4. 5-FU Plasma Concentration Quantification and Pharmacokinetic Profiling

The absorbance of 5-FU in rabbit’s plasma was tracked after the subcutaneous administration of free 5-FU solution and drug-loaded thermoresponsive gel formulations via already prepared calibration curve using HPLC system. A 5-FU estimation in the rabbit’s plasma from both receiving groups was determined using the Microsoft^®^ Office Excel 2013 program. Different pharmacokinetic parameters such as maximum plasma concentration (Cmax), time for maximum plasma concentration (T_max_), clearance (Cl), volume of distribution (Vd), elimination half-life (t_1/2_), area under the curve (AUC_0-t_), etc., were calculated through the scientific application package Kinetica^®^ version 5.1 (Thermo Electron Corporation) [35].

#### 2.16.5. Preliminary Safety and Histopathological Evaluation via Injectable Route

The selected chemically grafted Poly (NVCL-g-NaAlg) gel formulations were evaluated for acute toxicity studies in rabbits as per protocols of Food and drug administration authority (FDA, 2005). Maximal tolerance dose (MTD) for rabbits was calculated for toxicity evaluation via the subcutaneous route. Briefly, a gels sample containing 200–4000 mg/kg body weight 5-FU was injected through the subcutaneous route to six rabbits of 2–2.50 kg average body weight and monitored.

In order to conduct the study, twenty-four (24) rabbits with average body weight of 2.50 kg were selected and properly housed in a wooden box. The rabbits were, then, equally divided into three (3) groups; injectable gel formulation group (n = 8, 04 male and 04 female), positive control group (n = 8, 04 male and 04 female) and negative control group (n = 8, 04 male and 04 female). Before dosing, all rabbits were kept in a fasting state for 12 h, provided free access to water. The injectable gel group was administered 1000 mg/mL (total 4 mL) of 5-FU loaded formulation, while the positive control group was administered 1000 mg/mL (total 4 mL) pure free 5-FU solution through thesubcutaneous route, respectively. The negative group was injected equal volume of sterile saline. The rabbits were observed for behavior change, hair loss, feces, activity, mortality and body weight loss till two weeks. At end of the 2nd week, all rabbits were sacrificed and major organs such as liver, heart, spleen, kidney, and lung were subjected to gross histopathological examination for major pathological changes [35].

### 2.17. Statistical Analysis

Results are reported as percentage or mean ± standard deviation (SD). Statistical significance was tested by ANOVA using Graph Pad or Origin programs with *p* < 0.05.

## 3. Results and Discussion

### 3.1. Clarity of Formulations

The clarity of the in situ polymeric solutions and developed hydrogels reported were visually observed. All the samples’ solutions were found clear at all temperatures. The synthesized gels were also observed transparent at 4 °C, 25 °C, and 37 °C, indicating the solubility of all components. Table 2 indicates the clarity of developed formulations at different temperatures.

### 3.2. Solid State Characterization

#### 3.2.1. NMR Spectroscopic Analysis

To investigate the successful grafting between NVCL and NaAlg, pure monomer (NVCL), polymer (NaAlg), and the chemically grafted Poly (NVCL-g-NaAlg) gel samples were prepared in deuterium oxide (D_2_O) and analyzed by NMR spectroscopy. In the ^1^H-NMR spectrum of NVCL shown in Figure 1A, the peaks appeared in the range of 1.49–1.92 ppm corresponds to the protons for methylene group (-CH_2_- of caprolactam ring). The signals appeared at 2.29 and 3.51 ppm corresponds to the protons of methylene group (CH_2_) close to (N) and C=O of caprolactam ring, respectively. The signal appeared at 7.17 ppm corresponds to the proton of –CH= group, while signals appeared at 4.21 and 4.30 ppm are assigned to the two protons of the allyl group (CH_2_=), as shown in Figure 1A. The ^1^H-NMR spectrum of pure NaAlg shown in Figure 1B indicates the characteristic peaks in the range of 3.23–4.03 ppm which corresponds to proton positions of saccharide units in NaAlg. Moreover, the appearance of characteristic peak at 4.16 ppm refers to the hydroxyl proton peak of NaAlg. The ^1^H-NMR spectrum of chemically grafted Poly (NVCL-g-NaAlg) gel sample shown in Figure 1C indicates the evidence of demonstrating proton peaks from NaAlg (a) and from NVCL (b–e), which refers to the successful grafting and incorporation of NVCL on NaAlg backbone. As shown in the ^1^H-NMR spectrum of Poly (NVCL-g-NaAlg) sample, the shifted peak at 4.67 ppm corresponds to the hydroxyl proton peak of NaAlg. Similarly, the appearance of new peaks in the range of 1.45–2.46 ppm corresponds to the proton (H) positions for the methylene group (-CH_2_- of caprolactam ring), while characteristic peaks in the range of 3.08–3.71 ppm refers to the methylene groups (CH_2_) close to C=O and (N) of caprolactam ring, respectively. The appearance of characteristic new peaks in the ^1^H-NMR spectrum of Poly (NVCL-g-NaAlg) gel sample and the shifting of peaks refers to the successful chemical interaction and grafting between the NVCL and NaAlg moieties.

#### 3.2.2. FTIR Analysis

The structural changes of sodium alginate and N-vinylcaprolactam and the successful grafting of sodium alginate and N-vinylcaprolactam were confirmed by FTIR analysis.

Figure 2 shows the FTIR spectra of pure 5-FU, pure N-vinylcaprolactam (NVCL), pure sodium alginate (NaAlg), and chemically grafted Poly (NVCL-g-NaAlg) hydrogel (VCAlg). In the FTIR spectra of pure VCL, the characteristic carbonyl peak assigned to C=O stretching (amide I band) was observed at 1639 cm^−1^. The peak for the C=C was found at 1661 cm^−1^. The peaks in the range of 2931 cm^−1^ and 2863 cm^−1^ were assigned and correspond to the aliphatic C–H stretching. The characteristic -CH_2_- peak was observed at 1445 cm^−1^. The characteristic vinyl peaks (=CH and =CH_2_) were observed at 2992 cm^−1^ and 999 cm^−1^. The C–N stretching vibration peak was observed at 1482 cm^−1^. Furthermore, the characteristic broad peak at 3292 cm^−1^ is assigned to N–H stretching vibration, respectively.

The FTIR of pure NaAlg showed a broad peak at 3315 cm^−1^ assigned to the -OH stretching groups. The peak at 2393 cm^−1^ was assigned to –C–H stretching vibration. The peak at 1794.82 cm^−1^ indicates the -C=O stretching vibrations due to the presence of –COCH_3_ group. The peaks at 1549 cm^−1^ and 1387 cm^−1^ could be attributed to –CH_2_ scissoring and –OH bending vibrations, respectively. The peak at 1170 cm^−1^ suggested the presence of –CH–OH groups. The peak at 1017 cm^−1^ is assigned to –CH–O–CH- stretching.

In the FTIR spectra of VCAlg, the amide bands in the range of 1618 cm^−1^ and 1639 cm^−1^ get split due to the possible chemical interaction between VCL and NaAlg. The characteristic broad C=O peak observed between 1744 cm^−1^ and 2225 cm^−1^ is likely to be created by the formation of new hydrogen bonds between amide groups from VCL and carboxyl O–H groups of NaAlg. The occurrence of high-frequency shift of C=O in the range of 2539 cm^−1^ and 2871 cm^−1^ is suggested to be due to the destruction of H-bonds between NaAlg carboxyl groups, which corresponds to the involvement of carboxyl groups in H-bonding. The disappearance of the broad band of NaAlg in the FTIR spectra of VCAlg in the region of 3000–3600 cm^−1^ confirms the chemical reaction and formation of new bonding between amide and carboxyl groups.

#### 3.2.3. DSC Analysis

The DSC measurements of pure monomer, polymer, and grafted hydrogel sample were carried out under N_2_ atmosphere by heating in the temperature range of 25 °C to 300 °C.

The DSC thermograms of NVCL, NaAlg, and chemically grafted Poly (NVCL-*g*-NaAlg) hydrogel (VCAlg) are shown in Figure 3. The DSC curve of pure NaAlg displays a broad endothermic peak around 101 °C that is attributed to the loss of moisture contents of polysaccharide. A sharp exothermic peak at 240 °C was also observed in the DSC curve of NaAlg, which is assigned to the thermal decomposition of NaAlg at higher temperature. In the DSC curve of pure NVCL, initial endothermic or melting peak was observed in the range of 32–37 °C, which is suggested to be due to the hydration loss and hydrophobic interactions between carbon-carbon backbone chain of NVCL and, in turn, matches the phase transition above its LCST (32 °C). A second broad endothermic peak appeared in the range of 70 °C to 100 °C that indicates the decomposition of the thermoresponsive monomer at higher temperature. In the DSC curve of the chemically grafted hydrogel sample (VCAlg), first, a smaller endothermic peak was observed at 42 °C that is assumed to indicate the presence of VCL, followed by a broader endothermic peak in the range of 240 °C to 255 °C. This increase in the temperature of peak indicate that structural changes has occurred in the DSC of graft copolymer hydrogel by the introduction of NVCL. This alternatively refers to the formation of graft copolymer between NVCL and NaAlg.

#### 3.2.4. TG Analysis

The thermal stability of the pure NVCL, pure NaAlg and chemically grafted Poly (NVCL-*g*-NaAlg) in situ depot hydrogels was characterized by TG analysis from 20–300 °C under N_2_ atmosphere as illustrated in Figure 4. According to the thermogramms, NVCL has shown an initial abrupt weight loss at decomposition temperature (T_d_) around 110 °C suggested because of the liberation of absorbed moisture. Another degradation peak was observed in the thermogram of NVCL at decomposition temperature (T_d_) around 190 °C which indicates the complete weight loss and liberation of entrapped moisture. In the TG thermograms of pure NaAlg, initial weight loss starts at decomposition temperature between 60 °C and 200 °C associated with evaporation of the entrapped water followed by complete degradation. This complex process of degradation is suggested because of the dehydration of saccharide rings and breaking of C–O–C bonds in the chain of NaAlg. On the other hand, for chemically grafted Poly (NVCL-g-NaAlg) hydrogel sample (VCAlg), a smooth degradation was observed until 200 °C, followed by abrupt weight loss. However, a decrease in % weight loss was also observed for VCAlg as shown in Figure 4. This behavior of VCAlg with temperature change indicates the improvement in the stability of the copolymer and confirms the successful chemical grafting between NVCL and NaAlg.

#### 3.2.5. SEM Analysis

The surface and cross-sectional morphologies of chemically grafted thermoresponsive Poly (NVCL-g-NaAlg) in situ depot hydrogels are shown in Figure 5. The surface morphologies of chemically grafted thermoresponsive Poly (VCL-g-NaAlg) in situ depot hydrogels showed a smooth texture as indicated in Figure 5A,B at various resolutions. The morphological analysis of the cross-section of hydrogel samples revealed a highly inter connected porous morphology as shown in Figure 5C–F at low and high resolutions. These pores are hold responsible for the absorption of physiological solutions, water swellibilty, and diffusion of solute (drug, 5-FU) in and out of the gel network. The porous nature of the hydrogel network is believed to highly affect the swelling ability, drug loading, and release capacity.

### 3.3. Phase Diagram Measurement and Mechanism of Gelation

The key parameters for thermoresponsive in situ formed depot gels at physiologic temperature are the gelation temperature (Tsol-gel) and gelation time (Tg). The tube titling method was used for the determination of gelation time (Tg) and gelation temperature (Tsol-gel).

The gelation temperatures and time for the chemically grafted Poly (NVCL-g-NaAlg) depot gels (VCAlg-1 to VCAlg-10) in the presence of crosslinking agent has also been reported in Table 1. In case of chemically grafted thermoresponsive gels, the gelation occurs following the mechanism as described. Briefly, the reactive hydroxyl groups from NaAlg that act as free radicals formed during the reaction then react with the double bond of the vinyl monomer in the presence of crosslinking agent that creates the covalent bond formation between the monomer and NaAlg. It was found from the results that by increasing the concentration of (VCAlg-1 to VCAlg-3) crosslinking agent (0.75 mol % to 1.5 mol %), keeping the monomer and polymer concentrations constant, the gelation time reduces, while it has no significant effect on gelation temperature. This is because, the increasing concentration of crosslinking agent leads to faster reaction and consumption of functional groups in the reaction mixture. This leads to the covalent bond formation that result in increased cross-linked density and shorter gelation time. Moreover, for samples with increasing concentration of NaAlg (VCAlg-4 to VCAlg-7), similar gelation mechanism and thermoresponsive behavior has been observed as for physically cross-linked depot complexes. For samples with increasing concentration of NVCL (VCAlg-8 to VCAlg-10), gelation temperature and time were found to be decreased, which also followed the results of physically cross-linked depot complexes. This is believed to be due to the increased hydrophobic interactions of carbon-carbon backbone chain of NVCL. Figure 6 indicates the phase transition of chemically grafted thermoresponsive formulations.

### 3.4. Rheological Analysis

To ensure the usefulness of injectable hydrogels as drug delivery platforms, suitable mechanical strength and stability under physiological conditions are key factors to be considered. The rheological characteristics of an injectable hydrogel system directly affect the mixing capability of active pharmaceutical agent, its syringibility, and the release profile of the loaded pharmaceutical agent. To measure the mechanical strength and confirm the gelation of the injectable hydrogel at body temperature, a range of tests were applied to evaluate the values of storage modulus (G′) and loss modulus (G″), which defines the mechanical spectrum of the material.

#### 3.4.1. Time Sweep Test

The setting of chemically grafted Poly (NVCL-g-NaAlg) in situ depot hydrogels with time elution was investigated by monitoring their viscosity, G′, and G″ at 30 °C. For the phase transition of in situ formed depot hydrogels, the crossover point is considered a critical point where the G′ passes over G″ with time elution. From the time sweep data, it was observed that the G″ was greater initially for all the hydrogel samples which indicate the sol state of the samples. With time elution, the G′ increased faster than the initial G″ with a gelation point for each gel sample referring to the sol-gel transition. This phase transition point, where G′> G″ indicates the dominant elastic state of the samples, refers to the gel state of the samples. To investigate the effect of crosslinking agent (MBA) contents, the gelation kinetics was observed by monitoring the viscosity, G′, and G″ as function of time. Three samples (VCAlg-1, VCALg-2, and VCAlg-3) with varying MBA contents (0.75 mol %, 1 mol %, and 1.25 mol %) were prepared and subjected to dynamic rheology experiments. As presented in Figure 7, it was observed that the G′ increases with the increasing MBA contents in the feed composition of the gel samples. This higher G′ refers to the increased mechanical strength of the gel network. Moreover, the crossover points for these samples with increasing MBA contents were also observed to be decreased accordingly and found at 130 s, 110 s, and 102 s, respectively. This increase in G′ might be suggested of the increased cross-linked density that occurs because of increased consumption of the functional groups during polymerization reactions. Figure 7A–C indicates the oscillatory sweep tests and corresponding time data for chemically grafted depot gel formulations.

#### 3.4.2. Temperature Sweep Test

In the temperature ramp test, the values of G′ and G″ were evaluated over an extended temperature range as function of increasing temperature of circulating water bath attached to AR2000 rheometer. For chemically grafted Poly (NVCL-g-NaAlg) in situ depot hydrogels, temperature ramp test was conducted on selected samples (VCAlg-1, VCAlg-8, VCAlg-9, and VCAlg-10 containing 10%, 15%, 20%, and 25% of VCL, respectively) in order to look into their rheological profile. For chemically grafted gels, higher G″ values were observed in the start of the experiment, however, with increase in temperature, the hydrophobic interactions and aggregation of carbon-carbon backbone chain of NVCL dominated, which lead to sol-gel conversion of the formulations. This increase in G′ values induced by temperature change refers to the solid or gel state of the formulations. Figure 7D shows the change of G′ and G″ along with temperature for Poly (NVCL-g-NaAlg) gels.

#### 3.4.3. Frequency Sweep Test

The frequency sweep test describes the mechanical spectrum of injectable depot gel formulations which is monitored by observing the change in G′ and G″ as function of frequency change between 0.01–50 Hz under controlled strain of 1%. From frequency ramp test conducted for chemically grafted gel formulations, it was noted that G′ was found higher than G″ which refers to the stable structure of the injectable hydrogel formulations above the LCST. Figure 7E designates the change of G′ and G″ with variable frequency for gel formulations.

#### 3.4.4. Continuous Ramp Test

The thixotropic property of chemically grafted Poly (NVCL-g-NaAlg) in situ depot hydrogels was investigated by conducting continuous ramp test using AR rheometer 2000. The measurements were made as function of increasing shear rate between 0.1–10 s^−^^1^ for 10 min in flow mode. It was found that with increasing shear rate, all the formulations showed a progressive decrease in viscosity at room temperature. This indicates that gel formulations have shear thinning property which is suggested to be due to the disruption of chemical crosslinks along with increasing applied force between the groups. Figure 7F specifies the change in viscosity along with increasing shear rate.

### 3.5. Grafting Efficiency of Poly (NVCL-g-NaAlg) In Situ Depot Gels

Table 2 indicates the percent grafting efficiency (GE %) of chemically grafted Poly (NVCL-g-NaAlg) in situ depot gels. It was found that with increasing NaAlg and NVCL contents in the feed composition ratio of gels, % GE increased. This increase in % GE with increasing contents is suggested because of the presence of abundant functional groups in the reaction mixture. With increasing contents, more macro free radicals are generated in the propagation step that take part in the synthesis reaction. Since APS act as initiator and with increasing NaAlg and NVCL contents, APS attack the saccharide unit of NaAlg, which in leads to generation of more macro free radicals and active sites to react with the NVCL in the reaction mixture. This whole phenomena in turn leads to increased GE %.

### 3.6. Transparency Assessment of Formulations

The monitoring of change in optical transparencies of the chemically grafted formulations is another parameter to confirm the phase transition at physiologic temperature. The optical transmittances were observed with changing the temperature of digital water bath and measuring the transmittances using UV-Visible spectrophotometer (UV-1601 Shimadzu) at 450 nm. The measurements were made at variable temperatures below and above the LCST of the gel formulations. Figure 8A shows the change in formulation transparencies as a function of increasing temperature. It was found that in the start of experiment, the copolymer solution was transparent, homogeneous, and colorless. However, with temperature increase, their homogeneous nature changes rapidly as the temperature was increased gradually. It was found that with increasing temperature above the LCST, a sharp change in transparencies was observed for all the formulations which became quite opaque near 33 °C. Moreover, it was also noted that with increasing NaAlg and MBA contents, the change in transparencies also occurred rapidly. This rapid change is suggested because of the increasing cross-linked density of the gels. These measurements of the optical transmittances confirmed the LCST of the gels near around the body temperature.

### 3.7. Percent Crosslinking Determinations

MBA was used as crosslinking agent for the formulation of chemically grafted Poly (NVCL-g-NaAlg) in situ depot gels. It was noted that percent crosslinking increases with increasing contents of NaAlg, NVCL, and crosslinking agent. This is suggested because, the highest concentration of the polymers provides more functional groups during reaction polymerization. The presence of large number of functional groups increases the grafting reactions and in turn result in higher crosslinking yield and dense gel structure. Moreover, increasing crosslinking agent concentration also lead to increased percent crosslinking of the samples. This is because with the increase in crosslinking agent contents, most of the functional groups get consumed during the grafting reactions in the reaction mixture that alternatively lead to increased percent crosslinking. Figure 8B designates the effect of processing variables on crosslinking of Poly (NVCL-g-NaAlg) in situ depot gels.

### 3.8. In Vitro Degradation

In release of a bioactive drug, molecule-controlled degradation profile of delivery device is a key factor. The in vitro degradation profile demonstrated that chemically grafted Poly (NVCL-g-NaAlg) depot gels have a controlled degradation rate at 37 °C, as shown in Figure 8C. The weight loss for longer time period was shown by gel samples with higher polymeric contents (VCAlg-6) for approximately 8 days. The degradation profile of chemically grafted Poly (NVCL-*g*-NaAlg) depot gels demonstrated the characteristics of surface degradation, which mainly occur due to hydrolytic random polymer chains cutting. Such a mechanism offers prolonged delivery in a controlled fashion stable gels structure.

### 3.9. Swelling Experiments

The swelling studies of Poly (NVCL-g-NaAlg) depot gels were conducted in distilled water, pH 7.4 (PBS, 5 mM), and pH 1.2 at variable temperatures in closed containers. All the experiments were conducted in triplicates (n = 3). Swelling experiments were conducted to evaluate the effect of pH, temperature, crosslinking agent concentrations on swelling profile of gels.

#### 3.9.1. pH Sensitivity of the Chemically Grafted Poly (NVCL-g-NaAlg) Depot Gels

The pH dependent equilibrium swelling ratio of chemically grafted Poly (NVCL-g-NaAlg) depot gels was determined by conducting the swelling experiments in pH 7.4 (PBS, 5 mM) and pH 1.2 (0.1 M HCl). The highest swelling ratio for the chemically grafted gels were observed in pH 7.4 (PBS, 5 mM) at 25 °C. This is because of the fact that in basic media the carboxylic (–COO) groups from NaAlg remains in an ionized state that creates the chains’ repulsion and leads to swelling of the gel network.

Moreover, in acidic buffer solutions (pH 1.2), the ionized functional groups shift to deionized state. This deionization of the groups eliminates the repulsive forces from the gel network, which leads to the decrease in swelling ratio. However, in acidic solutions, the samples showed some extent of solvent uptake, which is believed to be due to the diffusion of the solvent into the hydrogel sample. Figure 9A denotes the response of chemically grafted Poly (NVCL-g-NaAlg) in situ depot gels to pH variations at constant temperature, i.e., 25 °C.

#### 3.9.2. Temperature Sweep Swelling Experiments

The response of the chemically grafted Poly (NVCL-g-NaAlg) in situ depot gels was determined by looking at the swelling kinetics, which indicates the response of swelling of hydrogel samples in response to certain temperature changes.

Temperature-dependent equilibrium swelling ratio was measured during temperature sweep experiments carried out at two different temperatures (25 °C and 37 °C) in variable swelling medias. It was observed that the in situ depot gels exhibit faster and highest swelling ratio at temperature lower than LCST of the hydrogel i.e., 25 °C. This is because of the fact that below LCST, the hydrophilic groups (-CONH-) of the VCL and NaAlg in the gel structure form an intermolecular entanglements with the surrounding water molecules, which leads to increased water uptake and SR. However, with the increase in temperature above the LCST (37 °C), the hydrogen bonds between water molecules and polymer network breaks. The water molecules changes from bound state to free state, which leads to elimination of the water molecules from the gel network and results in reduced swelling ratio. This response of the chemically grafted Poly (NVCL-g-NaAlg) in situ depot gels to temperature change was exhibited in all swelling medias. Figure 9B indicates the response of chemically grafted Poly (NVCL-g-NaAlg) in situ thermoresponsive depot gels to equilibrium swelling ratio at variable pH and temperature programs.

#### 3.9.3. Effect of MBA Concentration on Equilibrium Swelling Ratio (ESR)

In order to evaluate the effect of MBA concentration on ESR, three samples (VCAlg-1 to VCAlg-3) with varying MBA concentration were prepared and subjected to swelling experiments. The swelling experiments were carried out at 25 °C and 37 °C in pH 7.4 (PBS, 5 mM) and pH 1.2. It was noted that by increasing the concentration of MBA, ESR decreases accordingly. This is because, by increasing MBA, the pore size of the gel network decrease and the hydrogel structure become more compact. This compact structure formation results in decrease solvent molecules mobility inside the gel network which leads to decrease in ESR. Moreover, during polymerization reaction, most of the functional groups in the reaction mixture are consumed leaving behind low amount of free functional groups to bind with the solvent molecules. This alternatively leads to decrease solvent uptake and results in reduced ESR. This effect of increasing MBA concentration was observed at both temperatures and swelling medias. Figure 9C designates the effect of MBA on ESR as function of time while Figure 9D refers to the effect of MBA on ESR of chemically grafted Poly (NVCL-g-NaAlg) in situ depot gels at different pH and temperatures with variable contents.

#### 3.9.4. On–Off Switching Behavior

In order to investigate the temperature dependent swelling reversibility, oscillatory swelling tests (heating and cooling cycles) were performed. The oscillatory swelling cycles were performed for chemically grafted Poly (NVCL-g-NaAlg) in situ depot gels in phosphate buffer solution of pH 7.4 in two fan ovens set at different temperatures. Figure 9E display the on–off switching behavior of chemically grafted Poly (NVCL-g-NaAlg) in situ depot gels in PBS (pH = 7.4). The time interval between each cycle was set at 2 h and the reversibility process of response rate to temperature change was checked between 25 °C and 37 °C. Initially the unloaded xerogel disc was placed in buffer solution (pH 7.4, 5 mM) at 25 °C and after 2 h interval, it was relocated to buffer solution at 37 °C. It was observed that with the change of temperature, the hydrogel samples reversibly absorb and desorb solvent molecules that confirm the good reversible behavior of the hydrogels. This process was repeated several times. Since the time interval between each cycle was 2 h, the experimental data obtained and shown in Figure 9E are not the equilibrium swelling ratio; however, this data clearly indicate the swelling and deswelling behavior of the hydrogels occurs in 2 h interval.

### 3.10. Drug Contents

Selected chemically grafted gel samples containing 100 mg of 5-FU were incubated in extracted medium. All the samples were analyzed in triplicate for drug contents determination at 265 nm using UV-Visible spectrophotometer (UV-1601 Shimadzu, Japan). Table 2 indicates the drug contents recovered from the hydrogel samples. It was observed that with increasing crosslinking agent concentrations (VCAlg-2 to VCAlg-4), loading of drug contents decreased owing to the more compact hydrogel structure at higher crosslinking agent concentrations. Similarly, an increased in drug loading contents was observed with increasing NaAlg contents (VCAlg-5 to VCAlg-7). This is suggested because of the more hydrophilic gel structure and higher swelling due to the presence of abundant hydrophilic groups. On the other hand, with increasing NVCL contents (VCAlg-8 to VCAlg-10), drug loading was decreased owing to the thicken gel layer caused by more hydrophobic interactions.

### 3.11. Drug Release Study from In Situ Depot Gels

For the analysis of the drug release profile of chemically grafted Poly (NVCL-g-NaAlg) in situ depot gels, in vitro release tests were conducted in various dissolution medias, i.e., distilled water, pH 7.4 (PBS, 5 mM), and pH = 1.2 (0.1 M HCl). In vitro release experiments on selected samples were conducted to investigate the response of the in situ depot gels to various process parameters, i.e., pH, temperature of the dissolution media, polymeric contents, and degree of crosslinking.

#### 3.11.1. Effect of pH and Temperature on 5-FU Release

The pH of the surrounding medium is an important parameter that affect the release of encapsulated drug from depot gel matrix. The ionization of chemically grafted gels mainly depends upon the pH of the buffer solution as well as on the pKa values of the hydrogel components, which accelerate the release of encapsulated drug. To investigate the effect of pH of dissolution medium on 5-FU release, release experiments were conducted in pH 7.4 (PBS, 5 mM) and pH = 1.2 using USP dissolution apparatus-II (Pharmatest type PT-DT 7, Germany) at different temperatures. Cylindrical circular 5-FU loaded hydrogel discs (7 × 7 mm in dimensions) were immersed in respective dissolution media at specific temperature and the concentration of released drug was measured using UV-spectrophotometer. It was observed from the release profile that the hydrogel sample showed maximum drug release at pH = 7.4. This is because the carboxylate groups (–COO) in the gel network get ionized in basic media and creates the electrostatic repulsion between the network chains. These repulsive forces lead to chains’ relaxation that results in pores opening. These pores act as channels for diffusion of solvent molecules and release of the encapsulated drug from the gel matrix.

Alternatively, in acidic buffer solution (pH = 1.2), release of the drug from in situ depot gels decreased. This is because of the fact that in acidic environment, the carboxylate groups (–COO) become deionized leading to the elimination of repulsive forces from the gel network. This elimination of the forces results in decrease swelling and drug release. However, some of the drug release occur in acidic media that might be suggested due to the concentration gradient of drug and release by the uptake of some solvent molecules. The response of the chemically grafted depot gels to pH change was observed at both of the temperatures. Figure 10A indicates the response of chemically grafted thermoresponsive Poly (NVCL-g-NaAlg) in situ depot gels to pH variation of buffer solutions in terms of cumulative 5-FU release.

The mechanical properties as well as release behavior of thermoresponsive hydrogels changes with the change in temperature of the external environment. Thermoresponsive hydrogels (both negative and positive) have the ability to contract on heating or cooling above or below their LCST. This behavior of thermoresponsive hydrogels is suggested to be due to the increased hydrophobicity of polymer chains in hydrogel structure.

Effect of temperature on 5-FU release of chemically grafted thermoresponsive Poly (NVCL-g-NaAlg) in situ depot gels was investigated at two different temperatures (25 °C and 37 °C). The release experiments were conducted at pH 7.4 (PBS, 5 mM) and pH = 1.2, 0.1M HCl. It was observed from the release profile displayed in Figure 10A that release of the drug from the chemically grafted gels deceases with the increase in temperature. It was observed that the rate and amount of drug release was notably greater at 25 °C (swollen state) as compared to 37 °C (aggregated state). This is because, since the insitu depot gels contain NVCL in its side chains, above the LCST (37 °C), the hydrophobic interactions between the hydrophobic groups increase and lead to the aggregation of groups in the network structure. Due to the aggregation, the solvent uptake tendency of the gels deceases that results in reduced solvent uptake and drug diffusion in and out of the gel matrix.

In contrast, below LCST (25 °C), the hydrogel network become highly hydrated due to the increased interaction of water molecules with the network chains and results in increased water uptake. As a result of the increased hydrophilicity of the depot hydrogels, the entrapped drug molecules quickly diffuse out of the gel matrix. This in vitro release profile at different temperatures indicates that chemically grafted Poly (NVCL-g-NaAlg) in situ depot gels has good response rate and can be controlled by changing the stimuli such as temperature.

#### 3.11.2. Effect of NaAlg Contents on 5-FU Release

In order to evaluate the effect of NaAlg contents on 5-FU release, three formulations (VCAlg-5 to VCAlg-7) with increasing NaAlg contents from 1.25 wt% to 1.75 wt% were synthesized. The release profile of these formulations was investigated at pH 7.4 (PBS, 5 mM) and pH = 1.2, 0.1M HCl at two different temperature programs. It was observed from the invitro release profile that with increasing alginate contents in the feed composition of depot gels, release of the drug also increases. This is because NaAlg has integrated carboxylic groups (–COO) in its structure and with increasing its contents, the no’s of the carboxylic groups in the gel structure also increases. Due to the presence of large number of (–COO) groups, swelling and release of the drug also increases. As was discussed previously in swelling studies, in buffer solution of pH = 7.4 the carboxylic groups (–COO) converted into an ionized state (–COO-), which results in chains’ repulsions due to electrostatic forces and opening of the network pores. These pores opening results in faster and maximum drug diffusion out of the gel matrix in basic buffer solution. This response of the depot gels with increasing NaAlg contents was observed at both temperatures. However, the temperature dominant effect was observed in these experiments also.

Alternatively, in acidic buffer i.e., pH = 1.2, 0.1 M HCl, no significant drug release was observed due to the reduced ionization of the carboxylic groups (–COO) in acidic medium. However, an increase in drug release was also observed with increasing alginate contents. This drug release is suggested because of the hydrophilic nature of the in situ depot gels due to the presence of large number of carboxylic groups (–COO) in their side chains. Figure 10B–E indicates the effect of NaAlg contents in the feed composition ratio of chemically grafted Poly (NVCL-g-NaAlg) in situ depot gels in PBS (pH = 7.4) at different temperature programs as function of time and contents, respectively.

#### 3.11.3. Effect of Degree of Crosslinking on 5-FU Release

The release of the drug from the chemically grafted Poly (NVCL-g-NaAlg) in situ depot gels is governed by various physical and chemical parameters including those that have a direct connection with the release medium, release conditions (temperature and pH), preparation methods, and composition of the in situ gels. One other effective approach to alter the release rate of the gels is to amend the crosslinking density of the hydrogel matrix by using various concentrations of the crosslinking agent.

To study the effect of crosslinking agent concentration on drug release from the in situ depot gels, three samples (VCAlg-1 to VCAlg-3) with varying CA concentrations (1, 1.25 and 1.50 wt %) were synthesized and subjected to drug release experiments. The release experiments were conducted in DW and pH 7.4 (PBS, 5 mM) at a fixed amount of drug and polymeric contents as a function of two different temperatures. Figure 11A–C indicates the % cumulative release *vs* time profile and the effect of varying CA concentrations on drug release in buffer solutions of variable pH values and at different temperature programs. It was observed from the release profile that in situ depot gels showed a faster and higher amount of drug release at lower concentration of CA. However, with the increase in concentration, the cumulative release from the gels decreases and becomes slower. This is probably because of the fact that, at higher concentration of the CA, the hydrogel structure becomes more compact due to the contraction of the micro voids. As a result of this compaction, the hydrogel structure become more rigid and the free spaces in the network reduces that lead to lesser solvent penetration and diffusion of the drug out the gel matrix. The drug release studies were also supported by the swelling kinetics of the chemically grafted in situ depot gels prepared with varying concentrations of the crosslinking agent. Figure 11D refers to the effect of MBA (CA) on cumulative 5-FU release as function of variable contents at various pH and temperature values.

#### 3.11.4. Effect of N-Vinylcaprolactam (NVCL) Contents on Drug Release

To study the effect of N-Vinylcaprolactam (NVCL) contents on 5-FU release, three formulations (VCAlg-8, VCAlg-9 and VCAlg-10) with varying concentrations (15, 20 and 25 wt %) of Poly (N-Vinylcaprolactam) were synthesized. The release experiments were conducted in pH 7.4 (PBS, 5 mM) and acidic buffer (pH = 1.2, 0.1M HCl) as a function of two different temperatures of dissolution media.

It was noted from the release profile that drug release decreased with the increasing N-Vinylcaprolactam (NVCL) contents in the gel compositions. This is because NVCL has a hydrophobic carbon-carbon backbone chain and, with increasing its contents in the structure, the hydrophobic groups increases. This alternatively leads to highest hydrophobic interactions and results in aggregation of the network. This aggregation of the network structure then slows down the release of the drug. Figure 12A–D indicates the % cumulative release *vs* time profile and effect of varying NVCL concentrations on drug release in buffer solutions of variable pH values and at different temperature programs.

### 3.12. Networking Parameters of Poly (NVCL-g-NaAlg) In Situ Depot Gels

#### 3.12.1. Diffusion Coefficient (D)

Diffusion refers to the penetration of solvent or solute into the preexisting spaces in the porous network of the hydrogel sample. Diffusion coefficient of the chemically grafted thermoresponsive Poly (NVCL-g-NaAlg) in situ depot hydrogels were calculated in the phosphate buffer solution (pH = 7.4) owing to their highest swelling. Table 3 indicates the values of diffusion coefficient (D) along with increasing concentrations of MBA, NaAlg, and VCL. It was noted that diffusion coefficient (D) increases along with the increasing concentration of MBA (VCAlg-1, VCAlg-2, and VCAlg-3). This is because, with increasing MBA contents (crosslinking agent), swelling of the gel samples decreases owing to their compact structure. With the increase in NaAlg (VCAlg-5, VCAlg-6, and VCAlg-7) contents in the feed composition of hydrogels, values of D decrease, respectively, as shown in Table 3. This is suggested because of the higher water uptake of the gels with increasing NaAlg contents owing to the hydrophilic network structure. Moreover, it was also found that with the increasing NVCL (VCAlg-8, VCAlg-9, and VCAlg-10) contents in the feed composition of hydrogels, D values were found to be increased, accordingly. This may be suggested because of the hydrophobic nature of the NVCL, which thickens the gel layer with increasing its contents and hinders the water uptake and swelling.

#### 3.12.2. Molecular Weight between Crosslinks (Mc) and Solvent Interaction Parameters (χ)

The Mc values of chemically grafted thermoresponsive Poly (NVCL-g-NaAlg) in situ depot hydrogels were calculated by using Flory–Rehner theory. Table 3 indicates the values of Mc with increasing concentration of MBA, VCL, and NaAlg. A gradual decrease in the Mc values was noted with increasing MBA (VCAlg-1, VCAlg-2, and VCAlg-3) and NVCL (VCAlg-8, VCAlg-9, and VCAlg-10) contents in the feed composition of gel samples. Moreover, with increasing NaAlg contents, an increase in the Mc values was observed owing to the higher hydrophilic nature of gel sample as shown in Table 3.

Solvent interaction parameters (χ) of chemically grafted thermoresponsive Poly (NVCL-g-NaAlg) in situ depot hydrogels were calculated in buffer solution (pH = 7.4) to investigate the compatibility of hydrogel components with solvent. The (χ) values have inverse relationship with volume fraction of the gels. It was assumed that grater the values of (χ), weaker will be the interaction forces between copolymers in gels and fluids. It was noted from the results in Table 3 that with increasing MBA (VCAlg-1, VCAlg-2, and VCAlg-3) contents, χ values increases. This is suggested because of the growing cross-linked density that reduces the network pore diffusion or interaction of the solvent with structural components. It was also found that with increasing NaAlg (VCAlg-5, VCAlg-6, and VCAlg-7) and NVCL (VCAlg-8, VCAlg-9, and VCAlg-10) contents, χ values decreases accordingly. This behavior of gels refers to their hydrophilic nature and compatibility of the structural components with surrounding fluids. Moreover, NaAlg and NVCL are assumed to act as channeling agents in creating pores for the diffusion of solvent and solutes in the network structure.

#### 3.12.3. Polymer Volume Fraction

Polymer volume fraction (V_2,s_) is one other parameter that refers to the water uptake capability of chemically grafted thermoresponsive Poly (NVCL-g-NaAlg) in situ depot hydrogels. It was noted from the results in Table 3, that polymer volume fraction (V_2,s_) of gel samples increases with the increasing concentration of NaAlg (VCAlg-5, VCAlg-6, and VCAlg-7) and NVCL (VCAlg-8, VCAlg-9, and VCAlg-10). This indicates the hydrophilicty of gel samples and their increased solvent uptake ability. Moreover, it was also observed that polymer volume fraction decrease with increasing concentrations of MBA (VCAlg-1, VCAlg-2, and VCAlg-3) as shown in Table 3. This effect was suggested because of the compact network structure and highest cross-linked density that hinders the diffusion of solvent.

### 3.13. Drug Release Kinetics

Drug release kinetics and the release mechanism were predicted by fitting the in vitro release data obtained in various dissolution medias and variable temperatures to various mathematical. The release exponent (n), release constant (k), and regression coefficient (R^2^) obtained are listed in Table 4. It was observed from the results that sample (VCAlg-1) followed first order release kinetics in distilled water and pH = 7.4. Similarly, sample (VCAlg-2 and VCAlg-3) followed first order release kinetics in distilled water and zero order release kinetics in buffer solution of pH = 7.4, with their respective R^2^ values close to 1. This release kinetics of the gels is suggested because, the gels have highest swelling in phosphate buffer solution (7.4), and the release is suggested because of pore diffusion from gel network.

For chemically grafted thermoresponsive Poly (NVCL-g-NaAlg) in situ depot hydrogels, the release exponent (n) values were obtained by applying the Korsmeyer–Pappas equation to investigate the release mechanism. The “n” values were found higher than 0.5 for all samples in different dissolution medias as shown in Table 4 except for sample (VCAlg-7), which showed “n” values lower than 0.5. From release exponent value, it can be concluded that non-fickian diffusion could be the possible release mechanism which refers to the slow equilibrium release from depot gels above the LCST. However, for VCAlg-7, the release mechanism is suggested through fickian diffusion with “n” value lower than 0.5 which also involve the gel degradation along with diffusion.

### 3.14. In Vitro Cytocompatibility Study of In Situ Depot Hydrogels

MTT assay was used to evaluate the cytocompatibility of chemically grafted Poly (NVCL-g-NaAlg) in situ depot hydrogels against Vero cell lines. For cytocompatibility evaluation, two formulations of bare chemically cross-linked thermoresponsive Poly (NVCL-g-NaAlg) hydrogels (VCAlg-6 and VCAlg-10) with different concentrations were used. Cells treated with Triton X 100 were used as positive control. Cells devoid of the formulations) were used as negative control. Figure 13A shows the sketch of the cytocompatibility experiment. It is clear from the results that compared to the negative control (untreated cells), bare Poly (NVCL-g-NaAlg) in situ depot hydrogels (VCAlg-6 and VCAlg-10) also showed above 90% cell viability for all concentrations. However, cells treated with Triton-X 100 as positive control killed the cells which indicates the toxic nature of the compound. These results indicated that bare Poly (NVCL-g-NaAlg) in situ depot hydrogels were non-toxic to the Vero cell lines and can be used as in situ gel-forming controlled drug delivery depot system.

### 3.15. In Vitro Anticancer Activity of In Situ Depot Hydrogels

Figure 13B,C shows the plot which refers to the cells viability of free 5-FU solution and 5-FU loaded in Poly (NVCL-g-NaAlg) in situ depot hydrogels in wide concentration range (1, 5, 10, 20, 40, 60, 80, 100 μg/mL) against HeLa and MCF-7 cell lines. It was concluded from the results that 5-FU in solution form showed higher toxicity towards HeLa and MCF-7 cancer cell lines. It was also concluded that 5-FU retained its toxicity against the cancer cells even after incorporating in Poly (NVCL-g-NaAlg) in situ depot hydrogels (VCAlg-6 and VCAlg-10). It was observed that loaded depot hydrogels showed somewhat slower toxicity which confirmed the sustained anticancer activity of the hydrogels.

### 3.16. IC_50_ Values Evaluation

The IC_50_ values were calculated for free 5-FU and 5-FU loaded in gels towards HeLa and MCF-7 cancer cell lines from dose–response curves. The IC_50_ values for free 5-FU towards HeLa and MCF-7 were found to be 50.52 µg/mL and 53.58 µg/mL, while that for VCAlg-6 against HeLa cells was found around 39.91 µg/mL and 46.82 µg/mL against MCF-7 cells. Similarly, for VCAlg-10, the IC_50_ values were found to be 49.60 µg/mL against HeLa cells and 26.70 µg/mL against MCF-7 cells. These results showed that 5-FU loaded in Poly (NVCL-g-NaAlg) in situ depot hydrogels produced better cytotoxic activity than free 5-FU solution. Our results also confirmed that loaded Poly (NVCL-g-NaAlg) in situ depot hydrogels showed a lower IC_50_ value than free 5-FU against both the cancer cell lines, which refers to the better uptake and anticancer activity of in situ depot hydrogels. Table 5 indicates the IC_50_ values calculated for free 5-FU solution and 5-FU loaded in Poly (NVCL-g-NaAlg) in situ depot hydrogels.

### 3.17. In Vivo Absorption and Pharmacokinetic Profiling in Rabbits

The in vivo absorption of 5-FU was analyzed using albino rabbits (total, 24) as animal models and pharmacokinetic parameters were calculated. The plasma drug concentrations from each rabbits group were plotted as a function of time as shown in Figure 14. Table 6 refers to the pharmacokinetic parameters calculated after 5-FU injection in free and in situ loaded gel form at various time points, respectively. Various pharmacokinetics parameters were compared statistically with the reference formulation using student’s *t*-test with 95% (*p* < 0.05) confidence interval. The results of all pharmacokinetic parameters were found statistically significant at confidence interval with 95% (*p* < 0.05). Clear difference was noticed in plasma drug concentrations and all pharmacokinetic parameters of 5-FU administered in free solution form and depot gel form through the subcutaneous route. In current study the plasma drug concentration (C_max_) for 5-FU loaded in in situ gel form was significantly greater (VCAlg-6) in comparison to free 5-FU solution. The C_max_ for VCAlg-6 was 1433.599 ± 45.091 ng/mL within 36 h and, for reference, free 5-FU solution was found to be 2263.307 ± 13.36 ng/mL absorbed within 15 min, respectively. The higher C_max_ for VCAlg-6 with respect to time is suggested due to the slow release and absorption owing to the encapsulation of 5-FU in gel form in situ that remains intact and stays for the longer time at the absorption site releasing the encapsulated drug continuously at a constant rate. The rate of absorption after the subcutaneous administration of free 5-FU solution into the blood stream was rapid as depicted by a low T_max_ value, while for the in situ gel form the T_max_ was found significantly higher, which in turn indicates the slow absorption of drug from an in situ intact gel, indicating controlled release behavior. The T_max_ in case of free 5-FU solution was found to be 15 min, while, for VCAlg-6, it was extended to 36 h. The AUC_tot_ of 5-FU in pure solution form after injectable administration was calculated to be 26,630.03 ± 259.55 ng/mL × h and MRT values were found to be 14.19 ± 0.09 min, while for VCAlg-6, the AUC_tot_ was found to be 69,904.17 ± 1208.75 ng/mL × h, and MRT values were found to be 42.28 ± 0.68 h. The *p*-values were found statistically significant for all pharmacokinetic parameters. The higher values of all these parameters as compared to reference free 5-FU solution suggest the higher bioavailability of drug from in situ loaded gel form. The volume of distribution (Vd) and clearance (Cl) for VCAlg-6 was significantly high and low as compared to free 5-FU reference solution which in turn indicates the steady state drug concentration released from gel. The elimination half-life (t_1/2_) for VCAlg-6 and for free reference 5-FU solution was found to be 12.50 ± 0.74 h and 9.60 ± 0.00 min, respectively. The greater t_1/2_ for VCAlg-6 as compared to free solution suggests the slow elimination of drug from the body.

The plasma drug concentration time-plots shown in Figure 14A,B indicated that the drug loaded in situ depot gel (VCAlg-6) effectively retained the plasma 5-FU concentration for longer time duration after the subcutaneous administration with no significant toxicity. In addition, free 5-FU solution could not maintain the steady state concentration and rapidly cleared from the body in comparison to gel formulation. It was concluded from the in vivo analysis and pharmacokinetic parameters that in situ Poly (NVCL-g-NaAlg) depot gel formulations can be administered in vivo and can provide controlled drug delivery for longer time duration.

### 3.18. Tolerability and Preliminary Safety Evaluation

#### 3.18.1. General Conditions

No toxic effect was noted in any rabbit group injected with gel or positive and negative controls included in study during the whole study. No death of a rabbit in any group occurred while monitored in the whole 14-day period. The rabbits monitored displayed normal movement, eye slits, energy, behavior, hair, and teeth. The rabbits showed normal response to breathing, light, sound. Rabbits showed no dryness of mouth or nose, vomit or edema. Rabbits were monitored for normal feces in color, frequency, and lack of pus or blood. Food intake was equally found in all treated groups. Appendix A refers to the toxicity signs not found in all treated groups.

#### 3.18.2. Maximal Tolerance Dose (MTD)

MTD indicate the dose at which the object is continuously observed for toxicity, abnormality, morbidity or mortality. MTD refers to the highest dose accepted by animals which are under study for specific duration. So, animals were injected at 100–1000 mg/kg body weight dosing through the subcutaneous route. As no mortality or abnormal symptoms were seen at greater tested dose, MTD of 5-FU in Poly (NVCL-g-NaAlg) thermogels was established for 1000 mg/kg body weight via the subcutaneous route.

#### 3.18.3. Histopathological Examination

After the fourteenth day of the study, the samples collected from major organs of sacrificed rabbits were observed by light microscope for any pathological changes. It was noticed that 5-FU exhibited no significant pathological lesions after administration through the subcutaneous injection in free and loaded in Poly (NVCL-g-NaAlg) gels form.

#### 3.18.4. Major Organs

Figure 15 refers to the optical micrographs collected from major organs of different rabbits groups treated with gel formulations, pure 5-FU solution, and normal saline, respectively, through the subcutaneous injection. The micrographs of cardiac myocytes treated with pure 5-FU solution and thermogel formulations were found clear and arranged in good order with respect to the rabbit group treated with normal saline. Moreover, no necrosis or hemorrhage, inflammatory exudate, or any significant effect on pericardium, myocardium, or endocardium of the cardiac muscles was found. The microscopic images of liver shown in Figure 15 displayed no hepatocellar necrosis or degeneration. Moreover, no macrophages, lymphocytes, or neutrophils infiltration was noticed on the hepatic sinusoid. The spleen was found with normal spleen corpuscle having no gross pathologic changes in spleen sinus, red, or white pulp. The morphology of lungs of both treated groups displayed no bronchioles or alveolar epithelial denaturation. Moreover, the cells surrounding the bronchus showed no inflammatory infiltration. The airways cilia were found to be normal. The light microscopic images of rabbit’s kidneys of both treated groups and group treated with normal saline displayed that nephron shape is normal with defined shape around glomerulus. Moreover, there was found no necrosis, bleeding, or degeneration in glomeruli or the various other kidney tubes.

## 4. Conclusions

The current study reports the development of novel in situ forming chemically grafted formulations from N-vinylcaprolactam (NVCL) with sodium alginate (NaAlg) as a backbone using a combinations of the cold and free radical polymerizations methods. FTIR and NMR spectroscopy confirmed the successful grafting evidence of NVCL onto an NaAlg backbone. DSC and TGA analysis showed that developed formulations has phase transition within body temperature range and are thermally stable. SEM analysis confirmed that the developed thermoresponsive formulations has porous structure which assist in water and drug uptake and also facilitate the release of drug from gel network. Rheological analysis confirmed the viscoelastic properties and critical gelation temperature around 33 °C, while optical transmittance showed the temperature induced changes in formulations owing to the presence of thermoresponsive (NVCL) block. Swelling experiments and on–off switching behavior of the formulations at variable temperature programs confirmed that temperature has a dominant effect on the swelling kinetics. In vitro 5-FU release profile showed that release of 5-FU from in situ forming formulations was dependent upon the relaxation and aggregation of the gel network as function of changing medium temperature. Maximum 5-FU release was observed from gel formulations at PBS (pH = 7.4) and 25 °C owing to the presence of ionized carboxylic groups (COO-) in their structure and relaxed state. Results concluded that different network architecture components such as polymer, monomer, crosslinking agent, and drug have a significant effect on swelling kinetics and drug release profile of formulations. Results showed that porosity and mesh size calculated via the Flory–Rhener theory through investigating different structural parameters (D, Mc, V_2S_ and χ) significantly affect the swelling index and drug release behavior of formulations. Release kinetics suggested a variable release mechanism for different chemically grafted formulations. The in vitro degradation profile demonstrated that depot gels has controlled degradation rate. The safety of the formulations was confirmed through MTT assay against Vero cell lines. Results also showed that 5-FU entrapped in gel formulations induce controlled cell death tested against HeLa and MCF-7 cancer cell lines as compared to application of free 5-FU solution. Altogether, it was concluded that developed formulations has in situ crosslinking ability and can be used as a prolonged injectable drug delivery depot owing to their safe nature.

## Figures and Tables

**Figure 1 pharmaceutics-14-01050-f001:** ^1^H-NMR spectrum analysis of (**A**) pure N-vinylcaprolactam, (**B**) sodium alginate, and (**C**) Poly (NVCL-g-NaAlg) sample.

**Figure 2 pharmaceutics-14-01050-f002:** FTIR spectra’s of pure 5-FU, pure NVCL, pure NaAlg, and bare chemically grafted Poly (NVCL-g-NaAlg) in situ depot gel sample.

**Figure 3 pharmaceutics-14-01050-f003:** DSC analysis of pure NVCL, pure NaAlg, and bare chemically grafted Poly (NVCL-*g*-NaAlg) in situ depot gel sample.

**Figure 4 pharmaceutics-14-01050-f004:** Thermogravimetric analysis of pure NVCL, pure NaAlg, and bare chemically grafted Poly (NVCL-*g*-NaAlg) in situ depot gel sample.

**Figure 5 pharmaceutics-14-01050-f005:** SEM analysis of chemically grafted thermoresponsive Poly (NVCL-g-NaAlg) in situ depot hydrogels. Surface morphology ×100 (**A**), ×200 (**B**), Cross-sectional morphology ×200 (**C**), ×250 (**D**), ×300 (**E**), ×500 (**F**), ×800 (**G**), and ×1000 (**H**).

**Figure 6 pharmaceutics-14-01050-f006:** Sol-gel state transition of chemically grafted Poly (NVCL-g-NaAlg) depot gels.

**Figure 7 pharmaceutics-14-01050-f007:** Oscillatory sweep tests and corresponding time data for depot gel formulations under strain = 1% and frequency = 1Hz (**A**–**C**). Change of G′ and G″ over 25 °C to 40 °C with increasing temperature for chemically grafted Poly (NVCL-g-NaAlg) depot gels (**D**). Frequency sweep test of thermoresponsive chemically grafted Poly (NVCL-g-NaAlg) depot gels under controlled strain of 1% (**E**). Effect of increasing shear rate (0.1–10 s**^−^**^1^) on viscosity of gel formulations for 10 min at 25 °C (**F**). All the experiments were conducted in triplicates (n = 3).

**Figure 8 pharmaceutics-14-01050-f008:** Temperature dependence and effect of increasing polymeric and MBA contents on optical transmittances of thermoresponsive chemically grafted Poly (NVCL-g-NaAlg) depot gels (**A**) Effect of polymeric and crosslinking agent concentrations on percent crosslinking of the chemically grafted Poly (NVCL-g-NaAlg) depot gels (**B**). The data were analyzed for statistical significance with one-way ANOVA. The data were found statistically significant with *p*-value of <0.01. In vitro degradation profile of chemically grafted Poly (NVCL-g-NaAlg) depot gels at 37 °C (**C**). All the experiments were conducted in triplicates (n = 3).

**Figure 9 pharmaceutics-14-01050-f009:** The response of chemically grafted Poly (NVCL-g-NaAlg) in situ depot gels to swelling media of variable pH values at constant temperature i.e., 25 °C. A significant difference was seen in data at pH = 7.4 and 1.2. The data were analyzed with one-way ANOVA and found significant with *p*-value of <0.001 (**A**) Response of chemically grafted Poly (NVCL-g-NaAlg) in situ thermoresponsive depot gels to equilibrium swelling at variable pH and temperature programs (**B**) Effect of variable crosslinking agent concentrations (MBA) on equilibrium swelling ratio as function of time (**C**) Effect of MBA contents on ESR of chemically grafted Poly (NVCL-g-NaAlg) in situ depot gels at different pH and temperature programs (**D**). Heating and cooling cycles (swelling-deswelling-reswelling kinetics) of chemically grafted Poly (NVCL-g-NaAlg) depot gels (**E**) The data indicate the mean of (n = 3) individual experiments.

**Figure 10 pharmaceutics-14-01050-f010:** The response of thermoresponsive Poly (NVCL-g-NaAlg) depot gels to in vitro cumulative 5-FU release at variable pH and temperature programs. A significant difference was seen in drug release data at pH = 7.4 and 1.2. The data were analyzed with one-way ANOVA and found significant with *p*-value of <0.001 (**A**). Effect of increasing NaAlg concentrations in feed composition ratio of thermoresponsive Poly (NVCL-g-NaAlg) in situ depot gels on cumulative 5-FU release with time in PBS (pH = 7.4) at 25 °C (**B**), 37 °C (**C**), and in acidic buffer solution (pH = 1.2) (**D**). Cumulative 5-FU release in buffer solutions of variable pH values at different temperature programs with variable NaAlg contents (**E**). The data indicate the mean ± SD of (n = 3) individual experiments.

**Figure 11 pharmaceutics-14-01050-f011:** Effect of variable MBA contents on cumulative 5-FU release from Poly (NVCL-g-NaAlg) in situ depot gels with time in DW at 25 °C (**A**), in pH 7.4 (PBS, 5 mM) at 25 °C (**B**), and in PBS (pH = 7.4) at 37 °C (**C**). Cumulative 5-FU release in buffer solutions of variable pH values at different temperature programs with variable MBA contents (**D**). A significant decrease in drug release data was found with increasing MBA contents. The data were tested for statistical significance with one-way ANOVA and found significant with *p*-value of <0.1. The data show the mean ± SD of (n = 3).

**Figure 12 pharmaceutics-14-01050-f012:** Effect of NVCL contents on cumulative 5-FU release from Poly (NVCL-g-NaAlg) in situ depot gels with time in pH 7.4 (PBS, 5 mM) at 25 °C (**A**), pH 7.4 (PBS, 5 mM) at 37 °C (**B**), and in acidic buffer solution (pH = 1.2) (**C**). Cumulative 5-FU release in buffer solutions of variable pH values at different temperature programs with variable NVCL contents (**D**). The data indicate the mean ± SD of (n = 3).

**Figure 13 pharmaceutics-14-01050-f013:** In vitro cytocompatibility sketch of bare thermoresponsive chemically grafted Poly (NVCL-g-NaAlg) in situ depot gels against Vero cell lines using MTT assay (**A**). Cytotoxicity evaluation of chemically grafted Poly (NVCL-g-NaAlg) in situ depot gels against Human cervical (HeLa) cancer cell lines (**B**). Cytotoxicity evaluation of chemically grafted Poly (NVCL-g-NaAlg) in situ depot gels against MCF-7 cancer cell lines (**C**). Data reported show mean ± SD of (n = 3). The data were tested with one-way ANOVA and found significant with *p*-value of <0.01.

**Figure 14 pharmaceutics-14-01050-f014:** Plasma drug concentration-time plots of 5-FU administered through the subcutaneous route in free pure solution form (**A**) and loaded in depot gel form (**B**).

**Figure 15 pharmaceutics-14-01050-f015:** Histopathological micrographs of different rabbit organs of control, pure free 5-FU treated, and 5-FU loaded injectable gel groups after the subcutaneous administration.

**Table 1 pharmaceutics-14-01050-t001:** Feed composition of chemically grafted thermoresponsive Poly (NVCL-g-NaAlg) in situ depot hydrogels.

FormulationCodes	NVCL (g)	NaAlg (g)	MBA (g)	APS (g)	T_sol-gel_(°C)	GelationTime (T_g_)(Minute’s)	5-FU (mg)	Distilled Water(g)
VCAlg-1	1	0.100	0.075	0.150	35 ± 0.50	9	100	10 g
VCAlg-2	1	0.100	0.100	0.150	35 ± 0.60	8	100	10 g
VCAlg-3	1	0.100	0.125	0.150	35 ± 0.40	7.5	100	10 g
VCAlg-4	1	0.100	0.150	0.150	35 ± 0.50	6.5	100	10 g
VCAlg-5	1	0.125	0.100	0.150	35 ± 0.90	8	100	10 g
VCAlg-6	1	0.150	0.100	0.150	36 ± 0.20	9	100	10 g
VCAlg-7	1	0.175	0.100	0.150	36 ± 0.70	8.5	100	10 g
VCAlg-8	1.50	0.150	0.100	0.150	34 ± 0.70	6	100	10 g
VCAlg-9	2	0.150	0.100	0.150	33 ± 0.10	5	100	10 g
VCAlg-10	2.50	0.150	0.100	0.150	31 ± 0.90	5	100	10 g

**Table 2 pharmaceutics-14-01050-t002:** % crosslinking, % drug contents extracted, % grafting efficiency, and clarity of formulations.

Formulation Codes	Clarity of Formulations	% Crosslinking	Drug Contents %	% GraftingEfficiency
VCAlg-1	+++++	90.23	-	176
VCAlg-2	+++++	92.44	90 ± 0.38	189
VCAlg-3	+++++	95.69	87 ± 0.63	197
VCAlg-4	+++++	98.22	83 ± 0.22	205
VCAlg-5	+++++	88.84	92 ± 0.17	224
VCAlg-6	+++++	91.45	94 ± 0.12	237
VCAlg-7	+++++	93.72	95 ± 0.53	235
VCAlg-8	+++++	92.62	84 ± 0.18	193
VCAlg-9	++++	95.61	83 ± 0.33	228
VCAlg-10	++++	97.33	78 ± 0.79	208

**Clarity:** Good +++++, Fair ++++.

**Table 3 pharmaceutics-14-01050-t003:** Networking parameters of chemically grafted thermoresponsive Poly (VCL-g-NaAlg) in situ depot gels.

Formulation Codes	V_2,s_	χ	M_c_	D10^−6^ (cm^2^/s)
VCAlg-1	0.012	0.503	202,426.34	0.136
VCAlg-2	0.011	0.511	670,58.44	0.161
VCAlg-3	0.007	0.517	638,10.36	0.229
VCAlg-4	0.009	0.524	778,04.02	0.593
VCAlg-5	0.010	0.507	101,281.24	0.419
VCAlg-6	0.011	0.501	126,178.69	0.289
VCAlg-7	0.063	0.522	3592.22	0.416
VCAlg-8	0.092	0.533	1526.30	0.738
VCAlg-9	0.102	0.537	1230.47	0.815

**Table 4 pharmaceutics-14-01050-t004:** Drug release kinetics of chemically cross-linked thermoresponsive Poly (NVCL-g-NaAlg) in situ depot hydrogels at 35 °C.

Sample Codes	pH	Zero Order Kinetics	First Order Kinetics	Higuchi Model	Korsmeyer–Peppas Model
K_o_ (h^−1^)	R^2^	K_1_(h^−1^)	R^2^	K_2_ (h^−1^)	R^2^	n	R^2^
VCAlg-1	DW	3.840	0.994	0.0557	0.996	15.54	0.980	0.573	0.989
	7.4	3.108	0.985	0.0425	0.995	12.69	0.990	0.508	0.993
VCAlg-2	DW	3.673	0.990	0.0515	0.995	14.88	0.980	0.636	0.980
	7.4	3.901	0.998	0.0541	0.986	15.54	0.955	0.701	0.988
VCAlg-3	DW	3.025	0.979	0.0399	0.992	12.41	0.994	0.599	0.993
	7.4	3.325	0.996	0.0438	0.988	13.26	0.954	0.687	0.980
VCAlg-5	1.2	1.835	0.996	0.0213	0.998	7.395	0.974	0.591	0.983
	7.4	3.855	0.995	0.0557	0.984	15.42	0.961	0.574	0.984
VCAlg-6	1.2	1.700	0.966	0.0198	0.996	6.82	0.967	0.507	0.977
	7.4	3.969	0.991	0.0586	0.974	15.89	0.958	0.592	0.988
VCAlg-7	1.2	2.001	0.995	0.0241	0.992	8.081	0.963	0.482	0.973
	7.4	4.375	0.996	0.0671	0.991	17.63	0.975	0.579	0.988
VCAlg-8	1.2	1.694	0.994	0.0195	0.991	6.750	0.951	0.572	0.977
	7.4	3.681	0.996	0.0515	0.989	14.78	0.968	0.606	0.988
VCAlg-9	1.2	1.602	0.997	0.0182	0.996	6.419	0.965	0.593	0.985
	7.4	3.350	0.998	0.0443	0.989	13.34	0.954	0.645	0.981

DW = Distilled Water.

**Table 5 pharmaceutics-14-01050-t005:** The IC_50_ values and % Inhibition of pure 5-FU solution and 5-FU loaded in thermoresponsive Poly (NVCL-g-NaAlg) in situ depot hydrogels against HeLa and Breast (MCF-7) cancer cell lines and Vero cell lines.

SampleCodes	^a^IC_50_ (µg/mL) againstHeLa Cells	^a^IC_50_ (µg/mL) againstMCF-7 Cells	% Inhibition in VeroCells
Triton X100	-	-	81 ± 0.33
5-FU	50±0.52	53 ± 0.58	-
VCAlg-6	39±0.91	46 ± 0.82	6 ± 0.22
VCAlg-10	49±0.60	26 ± 0.70	9 ± 0.31

^a^IC50, concentration of 5-FU (or equivalent) (µg/mL) for inhibition of cellular growth by 50% after drug exposure for 24 h. Data reported are the percentages or mean ± standard deviation (n = 3).

**Table 6 pharmaceutics-14-01050-t006:** Pharmacokinetic parameters of pure free 5-FU solution and 5-FU loaded injectable hydrogels after the subcutaneous administration to healthy rabbits of Group-A and Group-B, respectively, (n = 24).

S. No.	Pharmacokinetic Parameters	Subcutaneous Pure 5-FU Solution(Mean ± SD)	5-FU Loaded Injectable Hydrogel(Mean ± SD)
1.	C_max_ (ng/mL)	2263.31 ± 13.36	1433.59 ± 45.09
2.	T_max_ (min)	15.00 ± 0.00	36.00 ± 0.00 (h)
3.	AUC_tot_ (ng/mL × h)	26,630.03 ± 259.55	69,904.17 ± 1208.75
4.	AUMC_tot_ (ng·h^2^/mL)	365,928.17 ± 3384.46	2.86 ± 0.07
5.	K_el_ (hr^−1^)	0.07 ± 0.00	0.09 ± 0.00
6.	t_1/2_ (min, h)	9.60 ± 0.00	12.50 ± 0.74 (h)
7.	MRT (min, h)	14.19 ± 0.09	42.28 ± 0.68 (h)
8.	Clearance (L/min)	0.73 ± 0.01	0.69 ± 1.42
9.	V_d_ (L)	2.97 ± 0.07	5.06 ± 0.29
10.	Vss (L)	10.34 ± 0.13	11.87 ± 0.13

## Data Availability

Yes, the data presented in the study are openly available.

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
