# Peer review of "RETRACTED: Poly (N-vinylcaprolactam-grafted-sodium alginate) Based Injectable pH/Thermo Responsive In Situ Forming Depot Hydrogels for Prolonged Controlled Anticancer Drug Delivery; In Vitro, In Vivo Characterization and Toxicity Evaluation"

_pharmaceutics, 2022, doi:10.3390/pharmaceutics14051050_

Round 1

Reviewer 1 Report

This manuscript by Khan and colleagues is a detailed analysis of the biocompatibility, drug delivery and biodegradability of the 5-FU containing Poly NVCL Alginate hydrogels. There is an extensive list of the necessary in vitro parameters required before the group proceeded towards preclinical model studies in albino rabbits. The results are of great interest to researchers studying biocompatible materials design, but require some changes to improve the readability:

  1. Minor formatting changes for consistency in writing °C and IC50 notations across Introduction and Materials sections
  2. Grammatical mistakes on Line 132
  3. Formatting error on Line 136, 420, 740, 831
  4. Wrong spelling of ‘Ununiform’ on Line 147
  5. Grammatical mistakes on Line 151
  6. Notation mistakes for chemical formula of ammonium persulphate on Line 204
  7. Clarity of gelation at different temperatures is only qualitatively compared, it is recommended that such reported parameters typically have a basis for their quantification as Good or Fair.
  8. Formatting of values in Table 2
  9. Statistical analysis in Figure 3B would be useful to compare significance of differences in crosslinking extent
  10. Inconsistencies in font size across the Results section
  11. Statistical analysis in Figure 6 panels would be useful to compare significance of differences in crosslinking extent affecting the drug release patterns
  12. Consistency of reporting of decimal figures in Table 3, 6
  13. Figures 4, 5, 6, and 7 are assembled such that the topmost parts of the panels are cut-off -- Please correct or re-prepare

Reviewer 2 Report

The article contains a consistent work related to injectable thermoresponsive in situ formed hydrogels based on poly(N-vinylcaprolactam-grafted-sodium alginate). The work is very consistent, using an important number of methods to characterize the synthesized hydrogels and also in vitro and in vivo testing, respectively. In my opinion, the paper is quite consistent and “bushy” and the obtained results could be the subject of two equally consistent papers. But, it remains to be seen what the other reviewers have to say... However, the paper may be published in this form, but with some “minor” modifications:

Abstract is too long; it contains 439 words (!). I recommend revising and reducing it to a maximum of 200 words (according to the Template)

The titles and subtitles are not properly marked in the text. This was done only in Introduction and Materials and Methods. After that, they are missing and it is very difficult to follow the text in this way.

References are not set according to the Template and need to be completely revised.

Line 136: “block” replaced with “block”

Line 144: …” 5-FU one of the best…” replaced with “5-FU is one of the best”

Line 147: “ununirom” refers to “ununiform”? If so, I recommend using “non-uniform”.

Line 214: “4°C, 25°C and 37°C” replaced with 4°C, 25°C and 37°C

Line 220: I recommend writing 90 degrees rather than “900

Line 268: Attention to the abbreviation of the measurement units such as minutes, hours ... the observation is valid for the whole text (!)

Line 277: Formula (1) and (2) are the same (!), so why are they marked differently?

Line 306: “…following formula was;” – something is missing here, to be revised.

Line 329: “A 10 ml of the 5-FU loaded sample (100 mg/ml) was cut…” – the sentence must be revised (How to cut a 10 ml sample?..., something is wrong…)

Line 427:  “pr°Cess” replaced with “process”

Line 431: “group B was injected 5-FU…” replaced with “group B was injected with 5-FU…”

Line 432: “…(2 ml containing 20 mg)” – 20 mg of …? I think it's about 5-FU, so it should be mentioned

Figure 1: should be replaced with a clearer figure, if the authors have this possibility, the gelling phenomenon is not very well observed.

Figure 2: abbreviate “seconds” with “s”; some words are Bold some are not; the figures do not have the same size.

Line 644: “is because as highest concentration of the polymers” – replace “as” with “the” – thus the sentence makes sense.

Line 645: “no” replaced with “number”

Line 654: “In release of a bioactive drug molecule or tissue engineering,..” – “tissue engineering” must be cut, it makes no sense to mention it considering the content of the paper

Figure 3: the percent of weight loss for all three tested gels is between 80 - 85%, so the statement from lines 657 and 658 “The maximum weight loss (%) was demonstrated by gel samples with higher 657 polymeric contents (VCAlg-6) in their feed compositions for approximately 8 days. “ is not justified. On the contrary, for VCAlg-4 the weight loss is faster (5 days) that VCAlg-6 (8 days). So the sentence must be revised.

Figure 4: the same font for the legend must be kept.

Line 711: It does not mention what ESR means? it is only mentioned below, at line 739. The first time a new term appears in the text; it must be written completely and then abbreviated.

Line 740: “°Ccur” – replaced with “occur”

Line 788: “were incubate…” – replaced with “were incubated”

Line 845: …”at pH=7.4, 5mM”… - it is PBS, so it should be mentioned after 5mM

Line 943: The authors often use the phrase "it was observed", sometimes twice in the same paragraph. Maybe the authors can find another variant of this expression...

Line 1068: I don't think it is necessary to present the “MTT Assay” again, because it has already been presented in the Materials and methods section.

Line 1090-1097: The presentation of the method for evaluating the anticancer potential of 5-FU should be moved to the Methods section.

Line 1166-1171: “5-FU was injected…using HPLC system.” - should be moved to the Methods section

Line 1294-1440: In my opinion, the instrumental analysis (NMR, FTIR, DSC, TG, SEM analysis etc.)  should be presented at the beginning of the results discussion (first characterization, then evaluation) and a little more brief and concise, to reduce a little of the” bushy” nature of the work…

Line 1442: Conclusions: I think they can be grouped in two distinct paragraphs, namely (i) the results of the gels characterization and (ii) the results after the gels evaluations, respectively, in order to be more easily followed by the readers.

After completing the paper, I would recommend changing the title to "... pH- and thermo-responsive in situ forming depot hydrogels ...", given that the synthesized hydrogels show an equally good response to changes in pH and temperature.

Reviewer 3 Report

Lines 89-94: In this section, more references are needed. The authors can take inspiration from these scientific works: https://doi.org/10.3390/jfb12020034; https://doi.org/10.1038/s41428-019-0217-0; https://doi.org/10.1016/j.carbpol.2021.118065.

Line 147: Probably, a typo is present.

Line 204: Chemical formula must be written following IUPAC nomencalture

Lines 260-262: How did author evaluate the “stability” by frequency sweep test? Stability in what sense?

Line 391: The used cell lines should be better described, as for breast (MCF-7) cancer cell line.

Lines 400-401: what were concentrations used for 5-FU?

Figure 1: The resolution of image is really bad and unprofessional

Line 582: Considering that temperature is an important parameter to evaluate and considering that other studies were carried out by the authors at room temperature and at 37°C, at what temperature was frequency sweep test carried out? If it was carried out only at room temperature, why did not use also body temperature?

Line 585: Why did authors choose these frequency values? a reference can be useful. Moreover, how did authors affirm that their samples are injectable? Without scientific evidence it is not possible to define the gel as injectable.

Lines 588-590: observing figure 2E, I don’t see “the change of storage modulus and loss modulus as function of variable frequency”. Moreover, considering that G’ showed greater values respect to G’’ for all range of frequency, it is very difficult to think of this gel as injectable. So, how did authors speck about INJECTABLE gel?

Lines 663-667: In this section, no results are described or discussed.

Lines 800-810: also In this case, no results are described or discussed. This part can be moved in material and methods section

Line 817: Why did authors choose these pH values? why such an acid pH? and why not also a basic pH?

Lines843-845: Considering that authors presented their samples as “thermoresponsive depot gel”, at 25°C the sample is gel form or in sol form? It is not clear and  more attention must be paid. If sample is in sol form at 25°C, an increased release of drug is obvious respect to a more structured matrix.

A big limit for the whole manuscript is the lack of statistical analysis, above all when different data are compared. Please compare the data by using a valid statistical analysis method and to show the results of statistical analysis in each figure.

Lines 921-923: why did authors use distilled water and PBS? What are the differences between the two solutions (of course apart of presence of salt)? What is the pH of used distilled water?

Lines 943-944: Please rephrase.

Lines 1045-1050: The sentences did not reflect the data reported in table. For example, VCalg-1 seemed to follow a first order release kinetics both in distilled water and pH 7.4. Please checked the data reported in table 4 and the data reported in the text.

Line 1076: Why were cytocompatibility studies carried out only on VCalg-6 and 10?

Lines 1103-1105: from in vitro studies, the effect of 5-FU loaded in the samples seems to be lower respect to free form. how can we take advantage of these results? we would have expected an increase in antitumor activity compared to the free drug

Moreover the data reported in table 5 seem to be not in line with the data reported in figure 8.

Round 2

Reviewer 3 Report

the authors answered all the questions. I have nothing else to add.

This manuscript is a resubmission of an earlier submission. The following is a list of the peer review reports and author responses from that submission.